# Continuous-variable fault-tolerant quantum computation under general noise

**Takaya Matsuura** [1,2] ✉, **Nicolas C. Menicucci**[1] ✉ **& Hayata Yamasaki** [3,4] ✉

Quantum error-correcting code in continuous-variable (CV) systems attracts much attention due to its flexibility and high resistance against specific noise. However, the theory of fault tolerance in CV systems is premature and lacks a general strategy to translate noise in CV systems into noise in logical qubits, leading to severe restrictions on correctable noise models. In this paper, we show that Markovian-type noise in CV systems is translated into Markovian-type noise in the logical qubits through the Gottesman-Kitaev-Preskill code. We analyze an upper bound on the resulting noise strength in terms of our newly introduced noise parameterization. Combined with the established threshold theorem of concatenated codes against Markovian-type noise, we show that CV quantum computation has a fault-tolerant threshold against general Markovian-type noise, closing the existing crucial gap in CV quantum computation. We also give a new insight into the fact that careful management of the energy of a state is required to achieve fault tolerance in CV systems.

Continuous-variable (CV) quantum optical systems, which encode quantum information into electromagnetic field quadratures, have distinct advantages for implementing quantum computation due to their affinity to quantum communication and the resulting scalability. There are established technologies for measuring the quadratures of an optical mode in the field of optical telecommunication. Furthermore, entangling operations are deterministic in the CV method, allowing the generation of a huge-scale entangled state even with current experimental technologies[1,2].

Fault tolerance is indispensable for a computation to be reliable, and quantum error correction (QEC)[3] is necessary to achieve fault tolerance. Various quantum error-correcting codes have been proposed for CV quantum computation[4–14] (see also ref. 15 for a review and comparison). Among others, the Gottesman–Kitaev–Preskill (GKP) code[9] has the advantage of easier implementation of the universal gate set and computational-basis measurement[9] as well as its error correction (EC) capability[15]. In fact, only in the preparation of the GKP state does one need non-Gaussian optical operations, which are difficult to perform in experiments[16,17]. The feasible generation of the (approximate) GKP state has been theoretically proposed in quantum

optical systems[18–21], and a primitive GKP state has been experimentally demonstrated very recently[22].

Despite these experimental progresses, however, the theory of fault tolerance in CV systems has not seen full maturity yet. In multi-qubit systems, how to achieve fault tolerance is well known and established[23–33]. On the contrary, fault tolerance of CV quantum computation has been shown only against specific noise models, such as Gaussian random displacement noise[34]. Many studies[35–40] have been claiming the existence of a fault-tolerant threshold of CV quantum computation with the GKP code following up the first study[34], but all of these analyses are against very restrictive noise models, such as a Gaussian-random displacement[41] and a Gaussian approximation of the GKP code[42]. Reference[37] proposed a twirling-like method that reduces a Gaussian-approximate GKP state to an ideal GKP state subjected to Gaussian random displacement noise, which may be applicable to other types of noise. However, this twirling-like operation cannot be physically realizable. Furthermore, virtually inserting a channel during the computation changes noise models, and therefore, this simplification cannot be used for a fault-tolerance analysis.

[1]Centre for Quantum Computation & Communication Technology, School of Science, RMIT University, Melbourne, VIC, Australia. [2]RIKEN Center for Quantum Computing (RQC), Wako, Saitama, Japan. [3]Department of Physics, Graduate School of Science, The University of Tokyo, Bunkyo-ku, Tokyo, Japan. [4]Department of Computer Science, Graduate School of Information Science and Technology, The University of Tokyo, Bunkyo-ku, Tokyo, Japan. ✉e-mail: takaya.matsuura@riken.jp; nicolas.menicucci@rmit.edu.au; hayata.yamasaki@gmail.com

Yet in experiments, indeed, there are non-Gaussian-type errors, such as a random phase rotation and experimental approximations of the GKP codeword proposed in refs. [18–21] and demonstrated in ref. [22]. These non-Gaussian errors may not be corrected perfectly by the EC and may pile up in the existing fault-tolerance analyses, leading to a breakdown in fault tolerance. Thus, the full fault-tolerance theory with the GKP code against general noise is indispensable for constructing a fault-tolerant optical CV quantum computer and guiding experimental efforts toward it.

In this paper, we close this crucial gap by showing a threshold theorem for optical quantum computation, regardless of the details of noise models. At first glance, most of the matured techniques developed in the fault-tolerance theory of qubit quantum computation[23–33] are not carried over to the GKP code since the ideal GKP codeword is nonnormalizable and thus not an element of the CV Hilbert space. Approximate codewords are valid quantum states, but their mutual non-orthogonality makes things more complicated. We show that by using the flexible framework to prove fault tolerance with concatenated codes[30–33,43], we can define a fault tolerance condition for the GKP code without relying on the unphysical ideal GKP codeword. The remaining question is whether (non-Gaussian) noise models on physical CV systems are translated to correctable qubit-level noise models or not via a concatenated code. We show that under mild assumptions on noise models in physical CV systems, the translated qubit-level noise model is actually a correctable one. Our assumed noise models cover experimentally relevant ones, such as non-Gaussian approximate GKP states, optical loss, and finite resolution of homodyne detectors. In this way, a threshold theorem for CV quantum computation is proved.

This paper thus provides a complete fault-tolerant digitization procedure for a quantum continuous variable, presenting a pathway to bridge the aforementioned gap between the existing fault-tolerance theory and current as well as future experiments of CV quantum computing. The obtained threshold theorem will guide experimental efforts on how to improve systems to meet the fault-tolerance criteria. This opens up the possibilities of CV fault-tolerant quantum information processing in a noisy real-world environment, as well as pushes up our understanding of CV quantum systems.

## Results
### Setting
A fault-tolerant protocol for quantum computation aims to approximate the output probability distribution of an ideal quantum computer with a quantum computer consisting of noisy devices in the real world. The approximation error should be smaller than an arbitrary parameter $\epsilon$ in the total variation distance. Such a fault-tolerant protocol is obtained in multi-qubit systems using concatenated codes[30–33,43], but not in CV systems except for a very restrictive case[34] as discussed in the introduction.

In this paper, we prove a threshold theorem for CV quantum computation by regarding CV systems as the physical (level-0) layer of a concatenated code and qubits defined through the GKP code as the level-1 layer of the concatenation. We use the framework of a fault-tolerant protocol for concatenated codes[30–33,43] in which each state preparation, gate operation, and measurement in the level-$k$ layer of concatenation are replaced with fault-tolerant *gadgets* consisting of operations for level-$(k−1)$ layer code, which are then sandwiched by EC gadgets of level-$(k−1)$ layer code. In the same way, we consider replacing each state preparation, gate operation, and measurement in the level-1 layer of a qubit-concatenated code with GKP state-preparation gadgets, GKP gate gadgets, and GKP measurement gadgets, which are then sandwiched by GKP EC gadgets (see Fig. 1). In this way, errors in CV systems are transformed into qubit-level errors by the GKP code. Now, the problem is to identify the noise model experienced by the qubits in the level-1 layer under

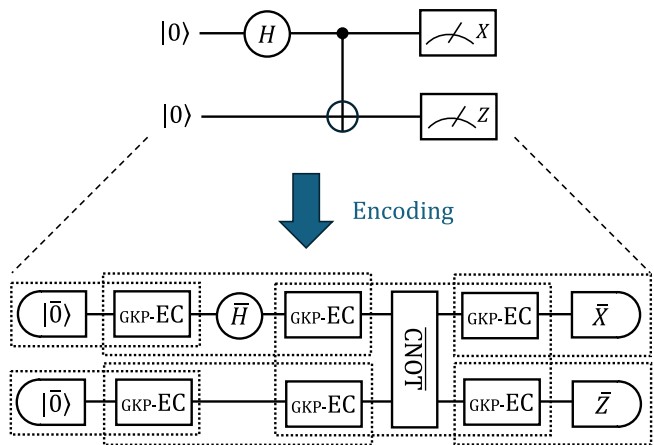

**Fig. 1 | A qubit quantum circuit replaced with (fault-tolerant) GKP gadgets with GKP error-correction (EC) gadgets inserted in between.** In our setup, the qubit circuit should be the level-1 layer of a concatenated code. Each dotted-lined box shows an extended rectangle (ExRec)[30], which is used in the level reduction theorem.

natural assumptions about noise in CV systems, and to determine whether it is correctable.

Reference [34] approached this question by assuming that the noise is Gaussian-random displacements, which we can handle relatively easily. Then, that work shows that the qubit-level noise will be a stochastic Pauli noise for this particular case. Note that even a Gaussian-approximate GKP codeword is not equivalent to the ideal GKP codeword subject to a Gaussian-random displacement noise. Extending this result immediately faces obstacles, which we describe below.

The first obstacle comes from the GKP code itself. The GKP code we treat here is a stabilizer code with the $2\sqrt{\pi}$-shift in position and momentum quadratures in the phase space being the stabilizer generators[9]. For the GKP codeword, which is invariant under these shifts, the $\sqrt{\pi}$-shifts in position and momentum quadratures act as logical Pauli-X and Z operators. Despite the conceptual clarity, the GKP code has an intrinsic problem: the ideal GKP codeword is not normalizable and thus is not a physically valid state. If one tries to define the ideal GKP state as a limit of the sequence of its approximation[9,42], then the energy (i.e., the average photon number) diverges in the limit. In fact, no Hilbert subspace is invariant under $2\sqrt{\pi}$-shifts in position and momentum. This appears to be a big obstacle since we no longer have a code space, which is always used in the analysis of an error-correcting code and in the theory of fault tolerance. The physically realizable state is only approximately $2\sqrt{\pi}$-shift invariant. This is the reason why the GKP code has been regarded as an approximate error-correcting code.

Another obstacle is the lack of distance measures between noise quantum channels that can deal with physically relevant situations while maintaining the properties that are necessary for fault-tolerance proof. In a finite-dimensional quantum system, the diamond norm is conventionally used as a distance measure, since it appears to be a relevant distance measure for quantum mechanics and satisfies the properties required for fault tolerance proof[29,44]. In an infinite-dimensional quantum system, however, this measure is too stringent: many physically relevant families of noise models have singular behavior with this distance measure[45,46]. For example, an infinitesimal phase rotation is not just far from the identity map but maximally distant from it. The reason for this counterintuitive behavior comes from the fact that quantum states with arbitrarily high susceptibility to a phase rotation have arbitrarily high energy (in this case, photon number). Since what we can generate in a lab does not have unbounded energy and thus such a state cannot be generated in practice (even with perfect experimental equipment), we need to use an alternative

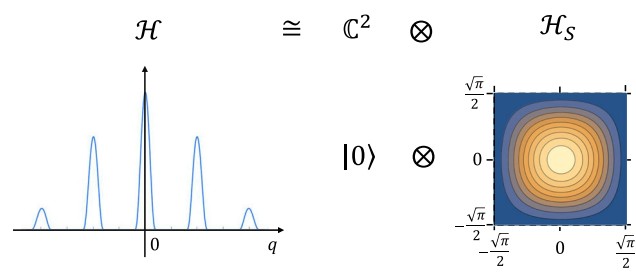

**Fig. 2 | The pictorial representation of the subsystem decomposition studied in refs. 47–51.** The above is a special case in which the state on the left-hand side can be written as a single tensor-product vector as on the right-hand side. A general pure state can be written as a linear combination of tensor-product vectors.

measure to reflect this experimental constraint. In the following, we explain how we overcome these obstacles and establish a fault-tolerance proof for CV quantum computation.

## CV fault-tolerance condition from stabilizer subsystem decomposition

The first obstacle we face is the unphysicality of the ideal GKP codeword and the resulting difficulty in defining a fault-tolerance condition. To circumvent this, we utilize a subsystem decomposition for the GKP code, first proposed in ref. 47 and elaborated in refs. 48–51. We will use the stabilizer subsystem decomposition[51], which remedied an undesirable asymmetry between the two quadratures that is present in the initial proposal[47]. It splits the Hilbert space $\mathcal{H}$ of the quantum harmonic oscillator into a tensor product of the Hilbert space $\mathbb{C}^2$ of the logical qubit and the infinite-dimensional Hilbert space $\mathcal{H}_S$ representing the syndromes of the stabilizer generators of the GKP code, i.e., $\mathcal{H} \cong \mathbb{C}^2 \otimes \mathcal{H}_S$ (see Fig. 2). In this decomposition, $\mathcal{H}_S$ is defined as the Hilbert space of square-integrable functions over the Cartesian square of a $\sqrt{\pi}$-sized interval[51]. Conceptually, this decomposition provides a CV generalization of the fact that a stabilizer code of $n$ physical qubits $\mathcal{H} = (\mathbb{C}^2)^{\otimes n}$ with a single logical qubit is decomposed into a tensor product of the Hilbert space $\mathbb{C}^2$ of the logical qubit and the Hilbert space $\mathcal{H}_S = (\mathbb{C}^2)^{\otimes n-1}$ of the $(n-1)$ syndrome qubits specified by the $(n-1)$ stabilizer generators, which can be represented as $\mathcal{H} = \mathbb{C}^2 \otimes \mathcal{H}_S$ in the same way[30].

With this decomposition, we introduce an equivalence class of errors with respect to which we define a fault-tolerance condition as in refs. 30,33 for the qubit case. For this purpose, we define the stabilizer-subsystem (SSS) $r$-filter and the ideal GKP decoder, which is analogous to the $r$-filter and the ideal decoder defined for concatenated codes in ref. 30. For the decomposition $\mathcal{H} = \mathbb{C}^2 \otimes \mathcal{H}_S$, the SSS $r$-filter gives us a tool to discuss how much the wave function of a physical state of the GKP code deviates from the origin of the Cartesian square in the definition of $\mathcal{H}_S$ (representing no error; see Fig. 2) and the ideal GKP decoder tells us the state of the logical qubit $\mathbb{C}^2$ at a given location in a circuit. The SSS $r$-filter depicted in Fig. 3 is defined as a projection operator acting as identity on the logical qubit while acting as a projection onto the sub-region $[-r, r) \times [-r, r)$ of the Cartesian square in the definition of $\mathcal{H}_S$ with $0 < r < \sqrt{\pi}/2$. The ideal GKP decoder, on the other hand, is a completely positive trace-preserving (CPTP) map that traces out $\mathcal{H}_S$ to transform a physical CV state of $\mathcal{H}$ into a state of the logical qubit $\mathbb{C}^2$.

Now, we can define an equivalence class of noisy or approximate GKP states, which we call the class of $r$-parameterized GKP states. An $r$-parameterized GKP state for a qubit state $|\psi\rangle$ is the product state of $|\psi\rangle$ in the logical qubit and a state that is supported only on the sub-region $[-r, r) \times [-r, r)$ in the Cartesian square in the definition of $\mathcal{H}_S$. Thus, the SSS $r$-filter acts trivially on it, and the ideal GKP decoder decodes it into $|\psi\rangle$. Unlike the unphysical ideal GKP codeword, an $r$-parameterized

GKP state is well-defined for any $r > 0$. Then, a fault-tolerance condition for this CV quantum computation is given such that a state preparation gadget prepares an $r$-parameterized GKP state with sufficiently small $r$, gate and measurement gadgets do not enlarge $r$ up to small constants, and an EC gadget refreshes a state to an $r$-parameterized GKP state while keeping the logical information.

Noticing that an $r$-parameterized GKP state can be written as a superposition of up-to-$r$ displaced ideal GKP states[47,51], it is necessary that gate and measurement operations do not amplify displacement errors to uncorrectable ones. This is analogous to the qubit fault-tolerance condition, where it is required that gate or measurement operations do not spread local errors. We show that the conventionally used gate and measurement operations for the GKP code[9] satisfy the fault-tolerance conditions defined above (see "Methods" and Supplementary Note 2B). It can also be shown that a gate followed by a small displacement error or a measurement after a small displacement error satisfies the fault-tolerance conditions. We can thus obtain a way to describe the fault-tolerance conditions for CV state preparations, gates, and measurements without referring to the details of noise and the shape of the wave function of an approximate GKP codeword. This is in sharp contrast to the previous work[34] where noise models and the shape of the wave function were explicitly assumed.

Now, we observe that an EC operation during computation has to get a state back to an $r$-parameterized GKP state with $r$ sufficiently smaller than $\sqrt{\pi}/2$, i.e., the threshold value of correctable displacement parameters for the GKP code. Otherwise, noise during computation accumulates and lets the support of the wave function in the Cartesian square of $\mathcal{H}_S$ go beyond $\sqrt{\pi}/2$, leading to a logical error. To get a state back into an $r$-parameterized GKP state, we need a GKP EC. However, we postpone the discussion of a GKP EC since the second obstacle—the need to take the energy constraint into account for obtaining an appropriate distance measure—also affects the requirements of a GKP EC.

## Energy-constraint condition

As explained so far, we have succeeded in abstracting the GKP code in a way that satisfies the requirements for fault tolerance. Thus, we can prove that a CV fault-tolerant circuit given as in Fig. 1, perfectly reproduces the outcomes of an original qubit circuit that we aim at simulating by the CV fault-tolerant circuit as long as all the CV circuit components perfectly satisfy the fault-tolerance conditions. However, this idealized situation will never be realized in experiments. We need tools to treat broader situations; the circuit components are almost perfect, but may have a general class of physical errors. This is where the distance measure for channels in CV systems is necessary.

As mentioned earlier, the diamond-norm distance, which is conventionally used in the fault-tolerance proof for qubit quantum computation[29,44], is too stringent for CV systems. In the literature[45,46,52–57], the energy-constrained version of the diamond norm has been considered, which induces a weaker and physically relevant topology of the set of quantum channels. The concept of imposing an energy constraint in the definition of the diamond norm was first introduced in ref. 55, but this definition is different from the one in refs. 45,46 in that ref. 55 constrains energy on both the input and reference systems. For the convenience of our analysis, we use the version in refs. 45,46 with the energy constraint only on the input system. For this, the mode-wise energy (i.e., the average photon number) of a state at every time step during computation needs to be bounded when averaged over measurement outcomes. Note that this is another reason why we need to avoid the reference of the ideal GKP codeword in the analysis, which has unbounded energy and may have an arbitrarily high susceptibility to phase rotation noise, which is numerically observed in ref. 51. Since CV gate operations, in general, increase the energy, a state during computation should continually be refreshed to a state with a constant energy upper bound.

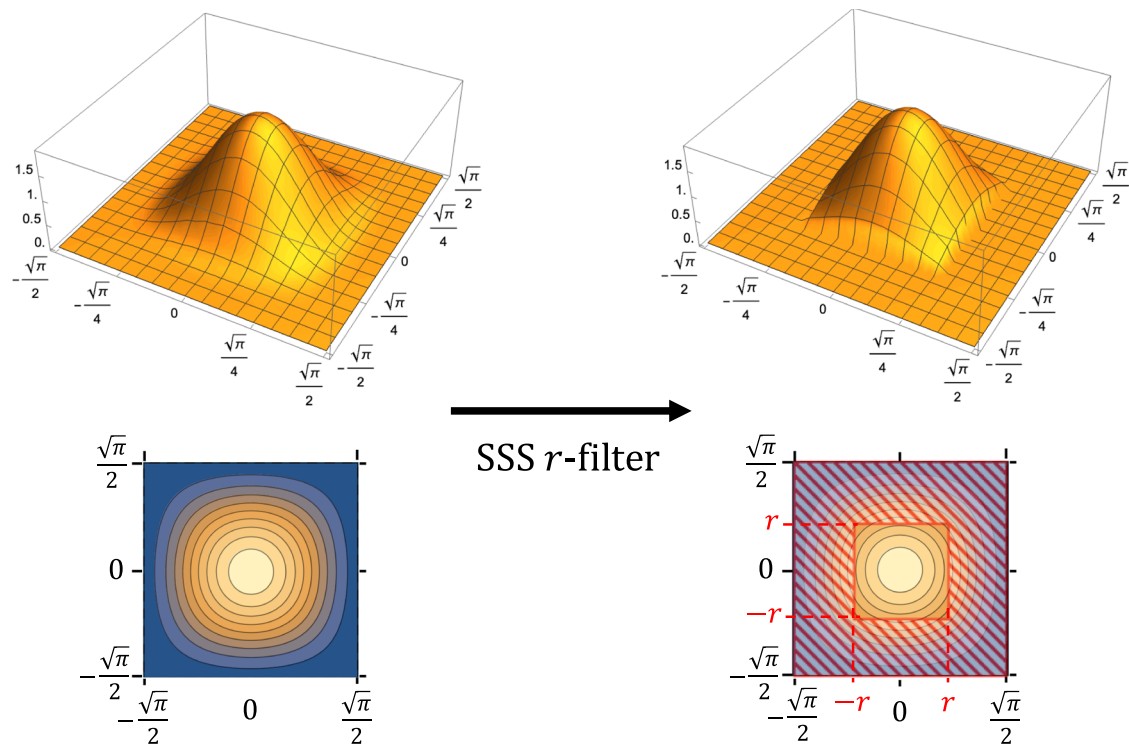

**Fig. 3 | The pictorial representation of $\mathcal{H}_s$ in the 3D plot (top) and the contour plot (bottom) with the action of the stabilizer subsystem (SSS) $r$-filter on it.** By the action of the SSS $r$-filter, the red-shaded region on the right-hand side is set to zero.

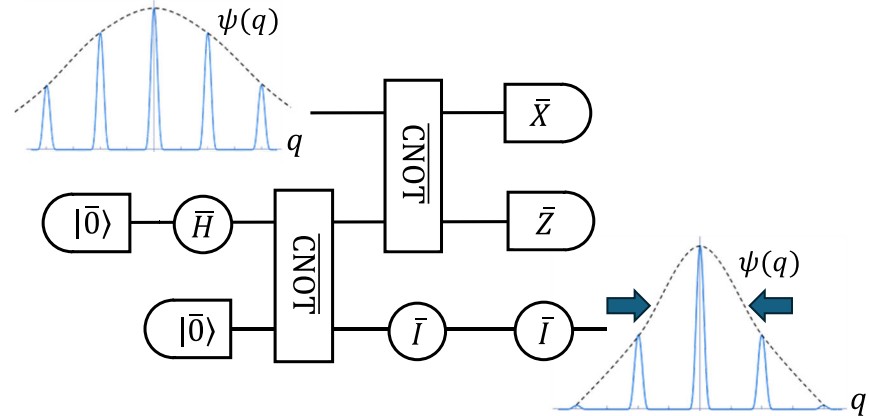

**Fig. 4 | Teleportation-based (Knill-type) GKP error correction[58] with the input and the output position wave function square.** In the figure, the feed-forward (GKP) Pauli correction operations are omitted; they can be corrected by the Pauli frame update or by modifying successive gates and measurements. Since a quantum state in the input is teleported to the ancillary prepared mode, the energy of a state is reset, which is depicted by the change of the variance of the wave function square. Note that the energy of a state also depends on the variance of the momentum wave function square.

To achieve this, we exploit a GKP EC gadget that satisfies both requirements of performing GKP EC and resetting the energy. This is accomplished by the teleportation-based (i.e., Knill-type) EC[58], as shown in Fig. 4, where the logical information during computation is teleported into a newly prepared auxiliary GKP codeword with a constant energy. It can be shown that if all the circuit elements in Fig. 4 satisfy the fault-tolerance conditions in terms of the SSS $r$-filter and the ideal GKP decoder, then this EC gadget also satisfies the fault-tolerance condition as a whole (see "Methods" and Supplementary Note 2B). Furthermore, as in Fig. 1, a state during computation is continually reset to a newly prepared GKP state while its logical information is preserved; this ensures that each prepared state during computation experiences only a constant number of gates before being measured in a GKP EC gadget. Then, we can prove that the energy of a state during computation has an upper bound (see Supplementary Note 2C).

This enables us to use the energy-constrained diamond-norm distance to assess the effect of noise. Unlike Knill-type EC, the conventional Steane-type GKP EC[9] may not reset the energy of an error-corrected state. Then, the energy of a state during computation may continue increasing and be more fragile against noise, which breaks down fault tolerance. Note that there may be ways to overcome this problem for Steane-type EC by a more complicated post-processing strategy, as implied in refs. 19,59,60, but it is difficult to evaluate an upper bound on the energy of a state during computation.

## CV FTQC under general noise

With the energy-constrained diamond norm, we introduce our noise model for CV systems, which covers experimentally relevant noise. Our noise model is independent in the sense that each mode experiences noise independent of that suffered by any other mode in the CV quantum computer, and it is Markovian in the sense that it has no correlation in time. Thus, we name this noise model an $(E, r, \epsilon)$-independent Markovian noise model, where $E$ is an energy constraint we take for the energy-constrained diamond norm, $r$ is a maximum amount of additional displacements to an ideal quantum operation, and a noise strength $\epsilon$ is the distance from this up-to-$r$-displaced ideal quantum operation measured in the $E$-constrained diamond norm. For state preparation, for example, $\epsilon$ is equal to the trace distance from an $r$-parameterized GKP state. This noise model not only covers non-Gaussian approximate GKP state preparation but also covers mode-wise optical loss, random phase rotation, and finite resolution of a homodyne detector. (See "Methods" for the formal definition.) This is in stark contrast to the previous work[34] where the noise model is restricted to Gaussian-random displacement.

Under this noise model, we aim to prove a threshold theorem for CV fault-tolerant quantum circuits. Our strategy to prove fault tolerance is based on the level reduction[30–33]. For this, we split a circuit into extended rectangles (ExRecs) in which each operation we want to implement is replaced with a fault-tolerant gadget, and ECs are inserted between these fault-tolerant gadgets. The ExRec is surrounded by the dotted line in Fig. 1. Each ExRec determines the behavior of the logical qubit-level circuit; i.e., if an ExRec is bad, the corresponding logical operation is regarded as erroneous. The detail of the level reduction is analyzed in "Methods" and Supplementary Note 2D. Combined with the threshold theorem for qubit quantum computation in ref. 31,32, we reach the following theorem. (See "Methods" for the sketch of the proof.)

**Theorem 1.** ((Informal statement of Supplementary Theorem 36 and Supplementary Corollary 37) Threshold theorem for CV quantum computation): We can construct a CV protocol using the GKP code in a way that for any target value $p > 0$, there exist nonzero thresholds $\epsilon_{th}$ and $r_{th}$ and a finite energy bound $E_{th}$ satisfying the following: even if each location of the physical CV circuit suffers from a general class of (not necessarily Gaussian) independent Markovian noise that deviates from its ideal action by (1) a distance parameter $\epsilon < \epsilon_{th}$, in terms of the energy-constraint diamond norm with energy $E_{th}$, and (2) a displacement parameter $r < r_{th}$, as in the noise model described above, it is guaranteed that the logical circuit of the GKP qubits undergoes local Markovian noise with noise strength below $p$. That is, by concatenating this CV protocol with a fault-tolerant protocol for a qubit concatenated code with a threshold $p_{th} > p$ under local Markovian noise, we can achieve fault-tolerant quantum computation using CV systems at the physical level with an overhead polylogarithmic in the size of an original circuit.

We remark that in the above theorem, the noise model that the logical circuit of the GKP qubits undergoes is no longer independent due to the correlation made by the EC procedures, but the occurrence of correlated noise is as rare as that for an independent noise model. This noise model, called local Markovian noise, is studied in detail for qubit systems in refs. 31,32 in which the threshold theorem is established. Thus, we can use this result to obtain the final statement of Theorem 1.

## Discussion

In this work, we rigorously showed the existence of a fault-tolerance threshold for quantum computation with the GKP quantum error correcting code under the general class of independent Markovian noise models, generalizing the result from ref. 34, which considered only Gaussian random displacement noise. Our noise model covers experimentally relevant noise, such as non-Gaussian approximation of the GKP states, optical loss, and finite resolution of homodyne detection. This can be proven using the stabilizer subsystem decomposition of the GKP code developed in ref. 51. The crucial difference from fault tolerance for qubit quantum computation is that the fault tolerance condition for the CV case requires (1) that displacements not be amplified by gate operations and (2) that the energy of a state during computation remains bounded. In contrast with the qubit case, without the energy condition (2), a CV state may become more and more susceptible to small noise and may eventually be uncorrectable. To avoid the accumulation of energy, as well as to correct CV-level errors, we use Knill-type EC[58] to ensure these conditions are satisfied. In practice, Steane-type GKP EC[9] will likely also work as long as some efforts are made to ensure energy bound on states[19]. Performing this analysis is left to future work.

We comment on how the use of analog information in the decoding affects our analysis, since the use of CV measurement outcomes often improves thresholds dramatically[35–37,39,40,61,62]. It is possible to gain nontrivial information from CV measurement outcomes only if some assumptions on the shape of a wave function on $\mathcal{H}_S$ is imposed. Existing analyses assume a Gaussian-like wave function on $\mathcal{H}_S$, which may no longer be justified for a general type of noise in our analysis. Thus, the use of analog information should be regarded as a noise-dependent heuristic rather than a general strategy to obtain a rigorous threshold, which we aim to achieve in this paper.

The implications obtained by the fault-tolerance framework and the threshold theorem developed here for CV quantum computation are many. First, the threshold theorem given here makes it clear what should be experimentally achieved in CV systems to build a CV quantum computer. As mentioned above, the noise model we treat in this paper is broad and contains a broad class of experimentally relevant noise. By estimating the noise strength $\epsilon$ of the actual experimental noise, one can judge that the noise level is low enough to implement a CV fault-tolerant quantum computer. Second, there may be a crucial difference between fault-tolerant quantum information processing with DV versus CV systems. In the CV case, the energy of a state during information processing should be carefully managed; otherwise, a state may become more and more fragile against noise. For the same reason, Knill-type EC and Steane-type EC may not be equivalent anymore in CV systems without careful adjustments. The former is ensured to reset the energy of the data mode, while the latter is not. These observations open up avenues for future studies on the possibilities and limitations of CV quantum computation and information processing.

## Methods

### The GKP code and the stabilizer subsystem decomposition

A harmonic oscillator is a prominent example of a CV system. A single-mode harmonic oscillator can be characterized by a pair of non-commutative quadrature operators $\hat{q}$ and $\hat{p}$ that satisfy $[\hat{q}, \hat{p}] = i$, acting on (a dense linear subspace of) a separable Hilbert space $\mathcal{H}$[63]. The energy level of a quantum harmonic oscillator is quantized and labeled by a non-negative number called the photon number. The photon-number operator $\hat{n}$ is defined as $\hat{n} := (\hat{q}^2 + \hat{p}^2 - 1)/2$.

We consider the GKP code for encoding a qubit into a harmonic oscillator. The ideal GKP-encoded state (the GKP codeword) $|\overline{\psi}\rangle$ is formally defined as an infinite superposition of position eigenstates, i.e., $|\overline{\psi}\rangle = \alpha |\overline{0}\rangle + \beta |\overline{1}\rangle$ with $|\overline{j}\rangle \propto \sum_{s \in \mathbb{Z}} |\sqrt{\pi}(2s + j)\rangle_q$ for $j = 0, 1$, where $|t\rangle_q$ satisfies $\hat{q}|t\rangle_q = t|t\rangle_q$ for the operator $\hat{q}$. The GKP code can be regarded as a stabilizer code with the stabilizer generated by $\exp(2\sqrt{\pi}i\hat{q})$ and $\exp(2\sqrt{\pi}i\hat{p})$. Logical Clifford unitary gates on this code can be realized by Gaussian unitaries, which are generated by second-

order polynomials of $\hat{q}$ and $\hat{p}$. The correspondence between the GKP logical operations and CV physical operations is given by ref. 9

$$\text{Pauli-X} : \overline{X} \to \exp(-\sqrt{\pi}i\hat{p}), \tag{1}$$

$$\text{Pauli-Z} : \overline{Z} \to \exp(\sqrt{\pi}i\hat{q}), \tag{2}$$

$$\text{Hadamard} : \overline{H} \to \hat{F} := \exp(\pi i\hat{n}/2), \tag{3}$$

$$\text{CNOT} : \overline{\text{CNOT}} \to \exp(-i\hat{q}_1\hat{p}_2), \tag{4}$$

$$\text{Wait} : \overline{I} \to \hat{I}. \tag{5}$$

Other logical gates can be implemented through gate teleportations as long as one can prepare GKP magic states[16,17,64]. More precisely, the logical phase gate $\overline{S}$ can be implemented through the gate teleportation with the state $|\overline{Y}\rangle := (|\overline{0}\rangle + i|\overline{1}\rangle)/\sqrt{2}$, and the logical $T$ gate $\overline{T}$ can be with the state $|\frac{\pi}{8}\rangle := (|\overline{0}\rangle + e^{\pi i/4}|\overline{1}\rangle)/\sqrt{2}$. Note that the CV shearing operation $\exp(i\hat{q}^2/2)$ is conventionally used to implement the logical phase gate $\overline{S}$ in the GKP code[9], but here we do not use this for simplicity of our analysis. The logical Pauli-X or -Z measurement in this code can be implemented by a homodyne detection in $\hat{p}$ or $\hat{q}$ quadrature, respectively, followed by the binning of the CV measurement outcome modulo $\sqrt{\pi}$. If the outcome is binned to an even (respectively odd) multiple of $\sqrt{\pi}$, it is regarded as logical 0 (respectively 1). This completes the description of how to perform a universal quantum computation with the GKP code.

The expression of $|\overline{\psi}\rangle$ above is only formal and does not represent a physically realizable state. For the code to be physically realizable, we need to approximate the state in some way[9], which complicates a theoretical treatment of the GKP code[42] and makes it difficult to analyze fault tolerance. It is also a problem of how or in what sense the state needs to be approximated, which we will resolve in this paper. The way of approximating the GKP codewords has been discussed in the literature[9,34,42]. The recent advance in this direction is the technique called the subsystem decomposition[47–51], which decomposes the Hilbert space of a CV system into a tensor product of the logical qubit and an infinite-dimensional Hilbert space.

The subsystem decomposition is similar to decomposing the physical Hilbert space of a stabilizer code into a tensor product of those representing the logical qubits and the syndrome qubits. In Steane's 7-qubit code[65], for example, the Hilbert space of the seven qubits can be represented as a tensor product of those of a logical qubit and six syndrome qubits on which the logical Pauli operators and the stabilizer operators, respectively, act nontrivially[30]. A codeword can be represented, up to this isomorphism, as a tensor product of any state of the logical qubit and a particular basis state of the syndrome qubits, say $|0\rangle^{\otimes 6}$ (i.e., the origin of the name syndrome qubits, representing the case of no error). A Pauli error on the physical qubits of a codeword flips some of the six $|0\rangle$s to $|1\rangle$s. The EC procedure for the 7-qubit code thus corresponds to detecting $|1\rangle$s in the syndrome qubits and correcting the overall state to $|0\rangle^{\otimes 6}$ of the syndrome qubits while acting as identity on the logical qubit. One may also regard the logical qubit and the syndrome qubits as subsystems in terms of this tensor-product decomposition as in refs. 66,67, where the subsystems in this context do not necessarily mean directly accessible ones, such as physical qubits, but are specifically defined from this tensor-product decomposition. In this sense, we may collectively call the syndrome qubits a syndrome subsystem for this tensor-product decomposition. All stabilizer codes can, in principle, be decomposed in this way, and so can the GKP code, where the Hilbert space of the syndrome qubits in the case of qubit stabilizer codes is replaced with an infinite-dimensional space of the syndrome subsystem for the GKP code. There are many different ways to define such a decomposition for the GKP code (called subsystem decomposition in the context of GKP codes[47–50]), and each corresponds to a different decoding strategy.

Among several ways of the subsystem decomposition developed so far for the GKP code, we utilize the stabilizer subsystem decomposition developed in ref. 51 because it corresponds to the typical binning decoder for GKP EC[9]. In the stabilizer subsystem decomposition, the Hilbert space $\mathcal{H}$ of a quantum harmonic oscillator is decomposed as

$$\mathcal{H} \cong \mathbb{C}^2 \otimes \mathcal{H}_S, \tag{6}$$

where $\mathbb{C}^2$ denotes the logical qubit and $\mathcal{H}_S$ denotes the Hilbert space of square-integrable functions over the square area

$$\text{Sq} := [-\sqrt{\pi}/2, \sqrt{\pi}/2) \times [-\sqrt{\pi}/2, \sqrt{\pi}/2), \tag{7}$$

as shown in Fig. 2. Thus, any state $|\phi\rangle \in \mathcal{H}$ can be represented as a linear combination of vectors of the form $|\mu\rangle \otimes \int_{(z_1,z_2)\in\text{Sq}} f(z_1, z_2)|z_1, z_2\rangle$, with $|\mu\rangle \in \mathbb{C}^2$ and $f \in \mathcal{H}_S$. We call this Hilbert space $\mathbb{C}^2 \otimes \mathcal{H}_S$ the SSS Hilbert space, and we call the former subsystem the logical qubit and the latter subsystem the syndrome subsystem in terms of this decomposition. (Note that the latter is originally called the stabilizer subsystem[51]). The GKP stabilizer operator $\exp(2\sqrt{\pi}i\hat{q})$ and $\exp(-2\sqrt{\pi}i\hat{p})$ acting on $\mathcal{H}$ can be represented on the SSS Hilbert space as $\hat{I} \otimes \exp(2\sqrt{\pi}i\hat{z}_1)$ and $\hat{I} \otimes \exp(-2\sqrt{\pi}i\hat{z}_2)$, respectively, where $\hat{z}_i$ for $i = 1, 2$ is an operator satisfying $\hat{z}_i|z_1, z_2\rangle = z_i|z_1, z_2\rangle$. The GKP logical Pauli-X operator $\overline{X}$ and the Z operator $\overline{Z}$ can be represented as $\hat{\sigma}_X \otimes \exp(-\sqrt{\pi}i\hat{z}_2)$ and $\hat{\sigma}_Z \otimes \exp(\sqrt{\pi}i\hat{z}_1)$, respectively, where $\hat{\sigma}_X$ and $\hat{\sigma}_Z$ denote the Pauli operators on a qubit. The ideal GKP state is a product of delta distributions in $z_1$ and $z_2$ and is thus an ill-defined state. However, it is clear from the above decomposition that the vector in the form $|0\rangle \otimes \int_{(z_1,z_2)\in\text{Sq}} f(z_1, z_2)|z_1, z_2\rangle$ has the same logical qubit state as the ideal GKP $|\overline{0}\rangle$ regardless of the form of the function $f$.

This decomposition can be derived from the fact that $\{\hat{q}\}_{\sqrt{\pi}}$ and $\{\hat{p}\}_{\sqrt{\pi}}$ commute with each other, while $\hat{q}$ and $\hat{p}$ do not[50], where $\{a\}_{\sqrt{\pi}} := a - \lfloor a \rceil_{\sqrt{\pi}}$ and $\lfloor a \rceil_{\sqrt{\pi}}$ denote the closest integer multiple of $\sqrt{\pi}$ for a real number $a$. Furthermore, each $\exp(\sqrt{\pi}i\lfloor\hat{q}\rceil_{\sqrt{\pi}})$ and $\exp(-\sqrt{\pi}i\lfloor\hat{p}\rceil_{\sqrt{\pi}})$ commute with both $\{\hat{q}\}_{\sqrt{\pi}}$ and $\{\hat{p}\}_{\sqrt{\pi}}$, while they two anti-commute with each other. In fact, $\exp(\sqrt{\pi}i\lfloor\hat{q}\rceil_{\sqrt{\pi}})$ and $\exp(-\sqrt{\pi}i\lfloor\hat{p}\rceil_{\sqrt{\pi}})$ acting on $\mathcal{H}$ correspond respectively to $\hat{\sigma}_Z \otimes \hat{I}$ and $\hat{\sigma}_X \otimes \hat{I}$ operators acting nontrivially on $\mathbb{C}^2$ of the SSS Hilbert space, while $\{\hat{q}\}_{\sqrt{\pi}}$ and $\{\hat{p}\}_{\sqrt{\pi}}$ acting on $\mathcal{H}$ correspond to $\hat{I} \otimes \hat{z}_1$ and $\hat{I} \otimes \hat{z}_2$ operators acting nontrivially on $\mathcal{H}_S$. In Supplementary Note 1A, we review the stabilizer subsystem decomposition in more detail.

## Fault-tolerant quantum computation and how to achieve it

The goal of fault-tolerant quantum computation (FTQC) is to construct a circuit $C'$ that simulates a $W$-qubit $D$-depth original quantum circuit $C$ within any given target error $\varepsilon > 0$; i.e., the circuit $C'$ outputs a $W$-bit string sampled from a probability distribution close to the output probability distribution of the original circuit $C$ within error in the total variation distance at most $\varepsilon$ when irrelevant measurement outcomes, such as those that appear in EC steps, are traced out[33]. Here, the depth means the number of time steps at which a preparation, gate, measurement, or wait operation is performed on each qubit. We call a physical operation performed on a given physical system at a given time step a location $C_i$ in the quantum circuit $C$, with $i \in \mathcal{I}$, where the index set $\mathcal{I}$ is chronologically ordered for simplicity of presentation

(among the locations at the same time step in $C$, the order is arbitrary). Note that $|\mathcal{I}| \leq WD$ holds by definition.

In qubit quantum computation, the achievability of fault tolerance has been shown[23-33]. This is made possible if an error on each circuit component is small enough and a physical circuit undergoes a reasonable noise model. For example, let $\mathcal{O}_i$ be an ideal map we want to implement at location $C_i$ in the circuit $C$, and let $\widetilde{\mathcal{O}}_i$ be its noisy version that we can implement in an experiment. If the noise is independent (meaning the noise has no space-like correlation) and Markovian (meaning the noise has no time-like correlation), then the noisy output distribution $p^{\text{noisy}}$ is given by

$$p^{\text{noisy}} = \widetilde{\mathcal{O}}_{|\mathcal{I}|} \circ \cdots \circ \widetilde{\mathcal{O}}_1 \tag{8}$$

$$= (\mathcal{O}_{|\mathcal{I}|} + \mathcal{F}_{|\mathcal{I}|}) \circ \cdots \circ (\mathcal{O}_1 + \mathcal{F}_1), \tag{9}$$

where the map $\mathcal{F}_i := \widetilde{\mathcal{O}}_i - \mathcal{O}_i$ is regarded as a fault. (Here, we assume that the set $\{C_i\}_{i \in \mathcal{I}}$ is chronologically ordered.) When the second expression above is expanded, each term except $O_{|\mathcal{I}|} \circ \cdots \circ O_1$ is called a fault path. The noise strength $\delta_i$ is then defined as the (diamond) norm of the fault $\mathcal{F}_i$ in the qubit case. If each $\delta_i$ is bounded from above by a constant $\delta$, then the total variation distance between the noisy output distribution $p^{\text{noisy}}$ and the ideal output distribution $p^{\text{ideal}}$ is bounded from above by $\varepsilon = (e-1)WD\delta$, where $e$ is Euler's number (see Supplementary Note 2D). To make the distance arbitrarily small, we need to make $\delta$ arbitrarily small, and for this, the circuit component should be encoded by a quantum error-correcting code. But again, the achievable noise strength is dominated by the noise strengths required to realize this quantum error-correcting code. As a result, one may need to use a code that can arbitrarily suppress errors by expanding the size of a code or by nesting a code over and over to achieve the target noise strength $\delta$. The latter choice is called a concatenated code[30-33], which we use in this paper.

When using a concatenated code, one needs to pay attention to how a noise model is translated from the physical level to the logical level. This is because the logical error may have correlations caused by the structure of a quantum error-correcting code and a fault-tolerant quantum circuit. The translation of the noise properties from the physical level to the logical level is studied in detail in refs. 30–33. In the case of independent Markovian noise at the physical level, the translated noise at the logical level is no longer independent Markovian noise. However, it is shown in refs. 31,32 that, at the logical level, the sum of all the fault paths with faults in $R$ specific locations is bounded from above by $\epsilon_{\text{qubit}}^{|R|}$ for a constant upper bound $\epsilon_{\text{qubit}}$ on the noise strength at any location, where $|R|$ denotes the cardinality of the set $R$. This noise model is called a local Markovian noise model. What is important is that if we further concatenate the qubit QEC code, then the local Markovian noise model in the lower-level qubit circuit is translated to local Markovian noise in the higher-level qubit circuit. Thus, there exists a threshold noise strength $\epsilon_{\text{qubit}}^{\star}$ below which we can arbitrarily suppress errors in a computational result. This whole process of reinterpreting a fault-tolerant quantum circuit at a given level with noise following a certain model as that at a higher level with noise following a translated noise model is called level reduction[30].

To establish this level reduction for a qubit concatenated code, refs. 30–33 introduced the concepts of the $r$-filter and the ideal decoder. The $r$-filter in these works is a projection operator onto a subspace $\mathcal{H}_r$ spanned by the vectors that are up to $r$ (Pauli) errors away from the codewords. Thus, for an error-correcting code that can correct up to $t$ Pauli errors (i.e., with code distance $2t + 1$), the $r$-filter defines an increasing sequence $(\mathcal{H}_r)_{r = 0, \ldots, t}$ of Hilbert subspaces with $\mathcal{H}_{r_1} \subseteq \mathcal{H}_{r_2}$ for $r_1 \leq r_2$, where $\mathcal{H}_0$ is the code space and $\mathcal{H}_t$ is the total Hilbert space of physical qubits that make up the code. The ideal decoder is a virtual quantum channel that performs a perfect decoding

operation (including EC) on the input state; that is, an error-correcting channel from physical qubits to logical qubits. The $r$-filter gives us a way to describe how far away given states are from the code space, while the ideal decoder tells us the state of the logical qubit at each time step. Using the $r$-filter and the ideal decoder, ref. 30 established fault-tolerant conditions for gadgets. For a qubit QEC code that can correct up to $t$ Pauli errors, a gadget that has at most $r$ faulty locations should output a state that is invariant under the action of $r$-filter and is decoded to the correct logical state via the ideal decoder if $r \leq t$. Then, a threshold theorem can be proved by relying only on these fault-tolerance conditions and the locality of noise, not relying on the detailed properties of noise. In this sense, the $r$-filter and the ideal decoder introduce an equivalence class of noise against which the fault-tolerant protocol works.

Following the idea of ref. 30, we establish a new level-reduction technique for CV FTQC. Since the GKP code is fundamentally different from qubit QEC codes, we need to introduce alternatives for the $r$-filter and the ideal decoder, which we name the SSS $r$-filter and the ideal GKP decoder. As the name suggests, we use the stabilizer subsystem decomposition explained in the previous section. For the SSS Hilbert space in Eq. (6), the SSS $r$-filter is defined as a projection operator that acts trivially on the logical qubit $\mathbb{C}^2$ and projects onto the subregion $[-r, r) \times [-r, r)$ of the square area Sq of the syndrome subsystem in Eq. (7) (see Fig. 3). The ideal GKP decoder is defined as a map that traces out $\mathcal{H}_S$ while leaving $\mathbb{C}^2$[51]. Thus, the SSS $r$-filter restricts the support of a wave function around the origin of the square area Sq of the syndrome subsystem, in contrast to the $r$-filter of the qubit concatenated code restricting the weight of Pauli errors. Again, the SSS $r$-filter defines an increasing net $(\mathcal{H}_r)_{r \in (0, \sqrt{\pi}/2]}$ of Hilbert subspaces $\mathcal{H}_r$ under inclusion. As $r$ gets smaller, all the states in the subspace $\mathcal{H}_r$ become better approximations of the ideal GKP states. There is, however, a mathematical subtlety for this net; the projective limit of this net seen as a projective system (i.e., $\bigcap_r \mathcal{H}_r$) is not the ill-defined ideal GKP code space in stark contrast to the qubit case. (In fact, $\bigcap_r \mathcal{H}_r = \{0\}$.)

The underlying principle of our definition of the SSS $r$-filter is that a large displacement error occurs less frequently than a small displacement error, even with faulty operations, which is intended in the construction of the GKP code[9]. See Supplementary Note 2B for the rigorous definitions. The SSS $r$-filter and the ideal GKP decoder are depicted as follows.

Here, the bold line denotes a CV system, and the thin line denotes a qubit. We will introduce the equivalence relation between noise through the SSS $r$-filter and the ideal GKP decoder as ref. 30 does for the qubit case. Let us define an $r$-parameterized GKP state $\hat{\rho}_\psi^r$ for a qubit state $|\psi\rangle$ as any state that is invariant under the action of the SSS $r$-filter and outputs $|\psi\rangle\langle\psi|$ under the action of the ideal GKP decoder. This state is a product state in the SSS Hilbert space: it is $|\psi\rangle$ in the logical qubit and has support only in the region $[-r, r) \times [-r, r)$ in the square area Sq of the syndrome subsystem in Eq. (7). Unlike the ideal GKP state, an $s$-parameterized GKP state can be a physically realizable quantum state. Let us further define an $s$-parameterized noise channel $\mathcal{N}^s$ as a noise channel whose Kraus operators are linear combinations of the elements of $\{\exp[i(u\hat{q} - v\hat{p})] : |u|, |v| < s\}$.

In the following, we first define classes of preparation, gate, measurement, and EC gadgets with parameters representing the spread of displacement errors in the gadgets. We then introduce fault-tolerance conditions for these gadgets, which essentially require that errors in the gadgets do not spread too much and the gadgets implement their intended logical operations if errors are correctable. The operations in Eqs. (1)–(4) satisfy these conditions. Finally, we

consider approximating general noise models with these fault-tolerant gadgets, which will be discussed in the next section.

The $s$-parameterized GKP preparation gadget ($s$-preparation in short) for a logical state $|\overline{\psi}\rangle$ is the preparation of an $s$-parameterized GKP state. The $s$-parameterized GKP gate gadget ($s$-gate in short) for a logical unitary $\overline{U}$ is the action of an ideal CV unitary $\hat{U}$ listed in Eqs. (1)–(4) followed by an $s$-parameterized noise channel $\mathcal{N}^s$. The $s$-parameterized GKP measurement gadget ($s$-measurement in short) is an $s$-parameterized noise channel $\mathcal{N}^s$ followed by an ideal homodyne detection and the binning of the outcome (see the main text). The $s$-parameterized GKP EC gadget is a Knill-type GKP EC circuit with all the circuit components replaced with $s_p$-preparations, $s_g$-gates, and $s_m$-measurements, where the parameter $s$ is a function of $s_p$, $s_g$, and $s_m$ (given below). Diagrammatically, they are written as follows:

$$\begin{array}{c}\boxed{|\overline{\psi}\rangle}^s \end{array} \longrightarrow \quad \hat{\rho}^s_\psi \quad , \tag{10}$$

$$\boxed{\overline{U}}^s \longrightarrow \boxed{\hat{U}}\boxed{\mathcal{N}^s} \quad , \tag{11}$$

$$\boxed{\overline{Z}}^s \longrightarrow \boxed{\mathcal{N}^s}\boxed{\hat{q}=t}\left\lfloor \frac{t}{\sqrt{\pi}}\right\rceil \bmod 2, \tag{12}$$

$$\boxed{\text{EC}}^{s'} = \begin{array}{c} \boxed{|\overline{0}\rangle}^{s_0}\ \widehat{H}^{s_H}\ {}^{s_\oplus} \boxed{\overline{X}}^{s_X} \\ \boxed{\overline{Z}}^{s_Z} \\ \boxed{|\overline{0}\rangle}^{s_0}\ \overline{I}^{s_I}\ \overline{I}^{s_I} \end{array}, \tag{13}$$

where the parameter $s'$ for the EC gadget is given by

$$s' := 2s_0 + s_H + s_\oplus + \max\{s_\oplus + \max\{s_X, s_Z\}, 2s_I\}. \tag{14}$$

Recall that we are not explicitly including the Pauli corrections since they result simply in a change of the Pauli frame that can be absorbed into future operations. Also note that the runtime of classical computation in the EC gadget (e.g., for binning of the measurement outcomes) can also be taken into account by running it during the wait operation in the EC gadget.

Now, with these definitions of fault-tolerant preparation, gate, measurement, and EC gadgets, we obtain fault-tolerance (FT) conditions for the SSS $r$-filter and the ideal GKP decoder. The fault-tolerant condition for an $s$-preparation is that a state stays invariant under the SSS $s$-filter and decodes to the ideal logical qubit state under the action of the ideal decoder if $s < \sqrt{\pi}/2$. The condition for an $s$-gate is that the output of the gate is invariant under the SSS $(r + s)$-filter with an $r$-filtered input state, and it decodes to the ideal logical qubit gate under the action of the ideal decoder if $r + s < \sqrt{\pi}/2$. The condition for an $s$-measurement is that, with an $r$-filtered input state, it decodes to the ideal logical qubit measurement under the action of the ideal decoder if $r + s < \sqrt{\pi}/2$. The condition for an $s$-EC is that the output state is invariant under the action of the $s$-filter and, with the $r$-filtered input, it decodes to the qubit identity map if $r + s < \sqrt{\pi}/2$.

It is important to stress that the satisfaction of the FT conditions for the gate gadget owes to the careful choice of the gate set in Eqs. (1)–(4). Here, the essential property that these gates have in common is that they do not enlarge displacement errors of an input (parameterized by $r$) up to a small additional constant $s$. They play the same role for displacement errors as transversal gates play for local errors. In this perspective, non-Gaussian gate operations, such

as $\exp(i\gamma\hat{q}^3)$ suggested in the original paper[9] to implement the GKP non-Clifford operation will make a displacement error destructively larger and cannot satisfy these FT conditions. Precise statements of the FT conditions and proofs are shown in Supplementary Note 2B.

An FT-GKP circuit $C'$ for a qubit circuit $C$ is constructed by replacing all locations $\{C_i\}_{i \in \mathcal{I}}$ in $C$ with respective FT-GKP gadgets, and the FT-GKP EC gadgets are inserted between these gadgets. To discuss the logical-level behavior of a given fault-tolerant circuit $C'$, ref. 30 introduced the concept of an ExRec, which we here generalize to the CV case. An ExRec is a part of the CV physical circuit composed of a gadget to implement logical operations of GKP qubits and the adjacent GKP EC gadgets, as illustrated in Fig. 1 in the main text. A preparation ExRec consists of a preparation gadget followed by an EC gadget; a gate ExRec consists of leading EC gadgets, a gate gadget, and trailing EC gadgets; and a measurement ExRec consists of an EC gadget followed by a measurement gadget. Although the meaning of the FT conditions in our setup introduced above is different from that of ref. 30, the equivalence relation of errors induced by these FT conditions is exactly the same form as that in ref. 30. Thus, the correctness condition for the GKP ExRec, i.e., the condition that its qubit-level circuit works ideally, is the same definition as that in ref. 30. Furthermore, we define that a GKP gate ExRec is good if it consists of a leading $s_1$-EC, an $s_2$-gate, and a trailing $s_3$-EC such that $s_1 + s_2 + s_3 < \sqrt{\pi}/2$. We define a good GKP preparation ExRec and a measurement ExRec in the same way. Then, with the same proof as that in ref. 30, we can show that a good GKP ExRec is correct. Diagrammatically, it states that

$$\boxed{\text{EC}}^{s_1}\ \boxed{\overline{U}}^{s_2}\ \boxed{\text{EC}}^{s_3}\rhd$$

$$= \boxed{\text{EC}}^{s_1}\rhd\ \bigcirc{U} \tag{15}$$

holds for a good GKP gate ExRec, where the thin circle denotes the ideal qubit unitary. Similar diagrammatic identities hold for a good GKP preparation ExRec and a measurement ExRec as well.

Unlike good GKP ExRecs, the logical qubit-level behavior of a bad GKP ExRec is not easily characterized. The reason is that the resulting logical error may depend on the CV state of the preceding ExRec. Thus, we cannot determine the erroneous behavior of a bad GKP ExRec solely with the given ExRec, but we need to see a larger context in the CV QC.

For this analysis, we introduce a new unitary operation called the GKP $*$-decoder, which generalizes the $*$-decoder for qubit concatenated codes[30]. This has a rather simple interpretation in the SSS representation: it is simply a unitary transformation mapping a state of a full mode to the equivalent state of the SSS Hilbert space $\mathbb{C}^2 \otimes \mathcal{H}_S$ in Eq. (6). This acts as a decoder because, in the SSS representation, the logical qubit is already what would result from applying ideal GKP EC. The GKP $*$-decoder faithfully reproduces all information in the original state because it keeps the syndrome subsystem $\mathcal{H}_S$ (instead of tracing it out as is done in the ideal GKP decoder). Diagrammatically, the GKP $*$-decoder is denoted as follows.

$$\rhd\quad \text{GKP } *\text{-decoder}$$

For a good GKP-gate ExRec, the GKP $*$-decoder makes the top thin wire into the action of the ideal unitary $U$, while the bottom thin wire into the action of a channel. Thus, the overall dynamics is decoupled between the logical qubit and the syndrome subsystem. For a bad

GKP-gate ExRec, we thus have the following.

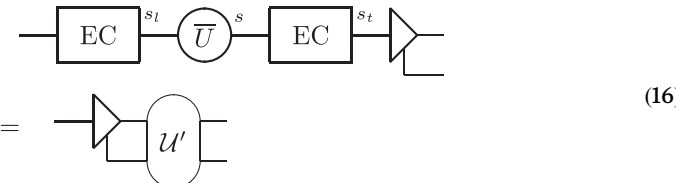

$$(16)$$

Here, $\mathcal{U}'$ denotes a channel that acts jointly on the logical qubit and the syndrome subsystem. Unlike Eq. (15) for a good GKP ExRec, we move the GKP *-decoder in Eq. (16) entirely to the left to remove the leading GKP ExRec for a bad GKP ExRec to avoid the noise correlation at the logical level. For details, see Supplementary Note 2D. Thus, the preceding GKP ExRec loses the trailing EC gadget. A GKP ExRec that loses the trailing EC gadget is called a truncated GKP ExRec. We also define the correctness and goodness/badness of a truncated GKP ExRec similarly.

Now, we can obtain a qubit circuit induced by the GKP code.

**Definition 2**. (Qubit circuit induced by the GKP code): Consider an FT-GKP circuit $C'$ on CV systems for implementing the original circuit $C$ on qubits. Then, a circuit $\widetilde{C}$ on qubits and environmental systems is defined by moving the GKP *-decoder from the end to the start of the circuit $C'$ as shown in Eq. (16) and regarding the syndrome subsystems as the environmental systems.

Recall that the GKP *-decoder transforms a good GKP ExRec for a location $C_i$ of $C$ to $C_i$ itself in $\widetilde{C}$ while independently changing the environmental system. Thus, if all the GKP ExRecs are good, they are correct, and all the locations $C_i$ of $C$ are included in $\widetilde{C}$ as it is. In this case, ignoring the environmental systems, we have $\widetilde{C} = C$ and obtain the desired outcomes with the same probability. Combined with the fact that each GKP ExRec has a constant number of locations, there exists a threshold value $s_{\text{th}}$ such that if the CV FT-GKP circuit $C'$ consists only of $s_{\text{p}}$-preparations, $s_{\text{g}}$-gates, and $s_{\text{m}}$-measurements with $s_{\text{p}}$, $s_{\text{g}}, s_{\text{m}} < s_{\text{th}}$, then all the GKP ExRecs in $C'$ are good, implying that $\widetilde{C} = C$.

However, the case where all the GKP ExRecs are good may be rare in the presence of noise. To achieve FTQC, we need to take into account how noise affects each location $C_i$ and makes some of the GKP ExRecs bad; even in this case, we need to prove that a fault-tolerant protocol can still correct errors occurring in bad GKP ExRecs by concatenating the GKP code and a qubit code. For this proof, we need to introduce an appropriate distance measure between CV quantum channels, as will be presented in the following.

## Energy constraint conditions

In the presence of noise on CV systems, it may still be too much burden to require that an actual experiment perfectly realizes $s$-preparations, $s$-gates, and $s$-measurements for sufficiently small $s$. Thus, here we consider noise models that cover general types of noise in CV systems and study how well the above conditions can be satisfied under the existence of such noise. Since we are mainly focusing on the optical system, and since the optical system does not repeatedly interact with the same environmental mode in typical setups, noise can be well modeled as Markovian; i.e., a noise map can be described as a quantum channel. Furthermore, we assume that modes used for CV quantum computation are well isolated so that they experience independent noise channels. These assumptions on noise models are standard in the theoretical analysis of fault tolerance, and generalization to more general cases, such as non-Markovian noise, may be possible by a modification of our analysis, as can be seen in the qubit case[31,32].

An immediate obstacle we face when we consider noisy physical operations on CV systems is the singular behavior of the diamond norm. The diamond norm $\| \cdot \|_\diamond$ has been conventionally used in the context of fault-tolerance theory since it has several properties

necessary to prove fault tolerance[26,29]. However, it is too stringent for noise models in CV systems, as can be seen in the following example. Consider a phase rotation channel $\mathcal{R}[\theta](\hat{\rho}) = e^{i\theta \hat{n}} \hat{\rho} e^{-i\theta \hat{n}}$. Then, one can show that $\| \mathcal{R}[\theta] - \text{Id} \|_\diamond = 2$ holds for any $\theta \in (0, 2\pi)$, where Id denotes the identity channel[45] since a coherent state with the amplitude $\alpha$ distinguishes these two channels better and better as $|\alpha| \to \infty$. This means that an infinitesimally small phase rotation is maximally distant from the identity map in terms of the diamond norm, even though such a small phase rotation is ubiquitous in experiments in practice. This also implies that we do not have any fault-tolerance threshold against phase rotation noise, however small it is, as long as it is measured by the diamond norm.

To avoid this limitation, a physically meaningful distance measure has been considered, which is named the energy-constrained diamond norm[45,46,52–57]. The idea of the energy-constrained diamond norm is not to consider the set of all states (which may contain unphysical states) but rather to consider the subset of states that have bounded energy. In the case of the quantum harmonic oscillator, this amounts to considering the set of states for which each one has an average photon number at most $E > 0$; then, the $E$-energy-constrained diamond norm takes maximization over this energy-constrained subset of states instead of maximization over the set of all states, denoted by $\| \Phi \|_\diamond^E$ for a Hermitian-preserving map $\Phi$. With this restriction of the set of states, we can avoid the singular behavior mentioned above[45,46].

We want to utilize this distance measure, but we need to check that the energy-constrained diamond norm actually satisfies the properties required for a fault-tolerance proof and that the energy of a local state during the computation has a constant upper bound. Whether the energy-constrained diamond norm satisfies the necessary properties for a fault-tolerance proof is nontrivial since—unlike the conventional diamond norm—a product of the energy-constrained diamond norms of Hermitian-preserving maps $\Phi$ and $\Psi$ is not an upper bound of the energy-constrained diamond norm of their composite map $\Phi \circ \Psi$. This is because the operator $\Psi(\hat{\rho})$ may not be a positive operator for a general Hermitian-preserving map $\Psi$, and the energy constraint for non-positive operators does not have a valid meaning. Nevertheless, we show that the following facts hold for any $E, E' > 0$ and the set $\mathfrak{S}_E(\mathcal{H}_Q)$ of energy-constrained states on a mode $Q$.

- For any Hermitian-preserving linear map $\Phi_Q$ and for any $\hat{\rho}_{QR}$ with $\hat{\rho}_Q \in \mathfrak{S}_E(\mathcal{H}_Q)$, we have $\| \Phi_Q \otimes \text{Id}_R(\hat{\rho}_{QR}) \|_1 \leq \| \Phi_Q \|_\diamond^E$. When $\mathcal{H}_R \cong \mathcal{H}_Q$, there exists a state $\hat{\rho}_{QR}$ that achieves the equality[53,68].
- For any CP map $\Phi : \mathfrak{S}_E(\mathcal{H}) \to \mathfrak{S}_{E'}(\mathcal{H})$ and for any Hermitian-preserving linear map $\Psi$, we have $\| \Psi \circ \Phi \|_\diamond^E \leq \| \Psi \|_\diamond^{E'} \| \Phi \|_\diamond^E$[45,46].
- For any Hermitian-preserving linear maps $\Phi_Q$ and $\Psi_{Q'}$, we have $\| \Phi_Q \otimes \Psi_{Q'} \|_\diamond^{E,E'} \geq \| \Phi_Q \|_\diamond^E \| \Psi_{Q'} \|_\diamond^{E'}$. The equality holds when at least one of $\Phi_Q$ and $\Psi_{Q'}$ is CP[45,46].
- Let two density operators $\hat{\rho}_{QR}$ and $\hat{\sigma}_{QR}$ satisfy $\hat{\rho}_Q, \hat{\sigma}_Q \in \mathfrak{S}_E(\mathcal{H}_Q)$ and $\| \hat{\rho}_{QR} - \hat{\sigma}_{QR} \|_1 \leq 2\epsilon$ with $0 < \epsilon < 1$. Then, for any Hermitian-preserving linear map $\Phi_Q$, we have $\| \Phi_Q \otimes \text{Id}_R(\hat{\rho}_{QR} - \hat{\sigma}_{QR}) \|_1 \leq 10\epsilon \| \Phi_Q \|_\diamond^{E/\epsilon^2}$.
- Let $\Phi$ be a Hermitian-preserving linear map and $\Psi, \widetilde{\Psi} : \mathfrak{S}_E(\mathcal{H}) \to \mathfrak{S}_{E'}(\mathcal{H})$ be two CPTP maps such that $\| \Psi - \widetilde{\Psi} \|_\diamond^E \leq 2\epsilon$. Then, we have $\| \Phi \circ (\Psi - \widetilde{\Psi}) \|_{diamond}^E \leq 10\epsilon \| \Phi \|_{diamond}^{E'/\epsilon^2}$.

The last two statements are our new results, which we prove in Supplementary Note 1C. Although properties listed here appear to be substantially weaker than those satisfied by the conventional diamond norm, we later see that these are sufficient to prove fault tolerance.

For the problem of the existence of a constant upper bound on the energy of a local state during computation, we impose that every preparation gadget generates a state with a constant energy bound $E_{\text{prep}}$ and every gate gadget increases the energy at most finite amounts as a positive, monotonically increasing, locally bounded function $g_{\text{sup}}(E)$ of the energy $E$ of an input state. See Supplementary Definitions 21 and 22 in Supplementary Note 2C for details. We assume that all the component gadgets in our fault-tolerant circuit satisfy the

energy-constraint conditions in the sense written above, which is in principle, testable in a real experiment. Then, as explained in the main text, using quantum teleportation, Knill-type EC continually resets the energy of a state. Note that we imposed the requirement that the energy of a state averaged over EC measurement outcomes should be bounded. This is because our ultimate goal is to show that the output probability distribution of this FT-GKP circuit $C'$ is close to that of the qubit circuit $C$, when measurement outcomes in EC steps are traced out.

Due to the energy reset by the Knill-type EC, there must exist a constant $\ell$ such that any state prepared during computation undergoes at most $\ell$ gates before being measured in a GKP EC gadget or at the final measurement of the CV circuit. Thus, a reduced state on each mode during computation has a constant energy upper bound $g_{\text{sup}}^{\ell}(E_{\text{prep}})$, where $g_{\text{sup}}^{m}$ for $m \in \{1,...,\ell\}$ is defined as

$$g_{\text{sup}}^{m} := \underbrace{g_{\text{sup}} \circ g_{\text{sup}} \circ \cdots \circ g_{\text{sup}}}_{m} . \tag{17}$$

See Supplementary Note 2C.

Having shown an upper bound on the local energy of a state during computation, we define the noise model we will be working with: $(s, \epsilon)$-independent Markovian noise model and $(E, s, \epsilon)$-independent Markovian noise model.

**Definition 3.** ((Informal version of Supplementary Definitions 8–10) $(s, \epsilon)$-independent and $(E, s, \epsilon)$-independent Markovian noise model): let $E_{\text{prep}}$ be a positive constant, and $g_{\text{sup}}$ be a fixed, positive, monotonically increasing, locally bounded function. Consider a collection $\mathcal{C}$ of physical systems that comprise a CV quantum computer, and regard other physical systems as environments. A noisy physical state preparation for the GKP logical state is said to obey the $(s, \epsilon)$-independent Markovian noise model if, independent of other physical systems in $\mathcal{C}$, it prepares a noisy state with the energy bound $E_{\text{prep}}$ that is $\epsilon$-close to an $s$-parameterized GKP state with the energy bound $E_{\text{prep}}$ in the trace distance. Furthermore, a noisy physical gate operation is said to obey the $(E, s, \epsilon)$-independent Markovian noise model if, independent of other physical systems in $\mathcal{C}$, it implements a CPTP map satisfying the $g_{\text{sup}}$-energy constraint that is $\epsilon$-close to a target unitary followed by an $s$-parameterized noise channel satisfying the $g_{\text{sup}}$-energy constraint in the $E$-energy-constrained diamond norm. Finally, a noisy measurement is said to obey the $(E, s, \epsilon)$-independent Markovian noise model if, independent of other physical systems in $\mathcal{C}$, it implements a CPTP map that is $\epsilon$-close to an $s$-parameterized noise channel followed by a target measurement in the $E$-energy-constrained diamond norm.

The formal statement is given in Supplementary Definitions 8–10 in Supplementary Note 1D. The above noise models cover physically relevant noise channels that we conventionally consider in CV quantum systems. In the subsequent section, we provide various such examples of the noise model, including conventional Gaussian approximation of the GKP codeword[9,42], random phase rotation, and a finite resolution of homodyne measurement. See also Supplementary Note 1D.

Now, we come to the point of stating our level-reduction theorem. A level-reduction theorem describes how physical noise models translate to logical ones through a fault-tolerant circuit and protocol. Let us choose parameters $s_{\text{p}}$, $s_{\text{g}}$, $s_{\text{m}}$, and $s_{\text{e}}$ sufficiently small so that GKP ExRecs consisting of $s_{\text{p}}$-preparations, $s_{\text{g}}$-gates, $s_{\text{m}}$-measurements, and $s_{\text{e}}$-ECs are always good and thus correct. Define $E_{\text{max}}^{\ell}(\epsilon)$ as

$$E_{\text{max}}^{\ell}(\epsilon) = g_{\text{sup}}\left(\frac{g_{\text{sup}}^{\ell-1}(E_{\text{prep}})}{\epsilon^2}\right), \tag{18}$$

where $\ell$ is again the maximum number of gates that a state prepared during computation undergoes. Then, we have the following.

**Theorem 4.** ((Informal statement of Supplementary Theorem 36) level reduction): consider an FT-GKP circuit $C'$ on CV systems for implementing the original circuit $C$ on qubits. Suppose that all the preparation gadgets in $C'$ satisfy the $E_{\text{prep}}$-energy constraint and all the gate gadgets in $C'$ satisfy the $g_{\text{sup}}$-energy constraint. Suppose further that the physical CV circuit $C'$ experiences the $(s_{\text{p}}, \epsilon)$-independent Markovian noise for state preparations, the $(E_{\text{max}}^{\ell}(\epsilon), s_{\text{g}}, \epsilon)$-independent Markovian noise for gates, and the $(E_{\text{max}}^{\ell}(\epsilon), s_{\text{m}}, \epsilon)$-independent Markovian noise for measurements, where $0 < \epsilon < 1$. Then, the logical-qubit circuit $\widetilde{C}$ implied from the FT-GKP circuit $C'$ by the procedure of Def. 2 undergoes a local Markovian noise with noise strength $\epsilon_{\text{qubit}}$ at each location satisfying

$$\epsilon_{\text{qubit}} = \mathcal{O}(\epsilon L_{\text{max}}), \tag{19}$$

where $L_{\text{max}}$ denotes the maximum number of GKP preparation, gate, and measurement gadgets in any truncated GKP ExRec of $C'$.

The proof is given in Supplementary Note 2D, with an upper bound on the coefficients of the $\mathcal{O}(\epsilon L_{\text{max}})$ term. Thus, we can immediately obtain a threshold noise strength $\epsilon_{\text{th}}$ in CV systems, given a threshold noise strength $\epsilon_{\text{qubit}}^{\star}$ against the local Markovian noise model for a qubit QEC code that is concatenated with the GKP code. Since a qubit concatenated code has been known to have a fault-tolerance threshold $\epsilon_{\text{qubit}}^{\star}$ against the local Markovian noise model[32], we reach Theorem 1 in the main text (see Supplementary Note 2 for details). In particular, ref. 32 shows that, by concatenating the Steane code, the noise strength at the $k$th level of concatenation decreases doubly exponentially as $\mathcal{O}(c^{2^k})$ with $c := \epsilon_{\text{qubit}}/\epsilon_{\text{qubit}}^{\star} < 1$ while the space and time overhead increases exponentially in $k$. Thus, a constant accuracy $\varepsilon$ for a logical outcome of $W$-qubit $D$-depth circuit can be achieved within $\mathcal{O}(\text{poly}(\log_{1/c}(WD/\varepsilon)))$ space and time overheads[32].

## How conventional CV noise models fit to our noise parameterization

We will demonstrate how conventional CV noise models can be interpreted as $(s, \epsilon)$- or $(E, s, \epsilon)$-independent Markovian noise models. For state preparation, a Gaussian-approximate GKP state $|\overline{0}_{\sigma^2}^{\text{app}}\rangle$ has conventionally been studied[9,42], where the variance $\sigma^2$ of each peak in the phase space is considered to characterize how good the approximation is (See Supplementary Note 1D for its explicit expression.) The ratio of the variance $\sigma^2$ and that of the vacuum is called the squeezing level in an analogy to a squeezed state and often represented in decibels. When one performs the stabilizer subsystem decomposition on this state, one can see that it has nonzero support all over the region Sq defined in Eq. (7). Thus, it is not an $s$-parameterized GKP state, and we should interpret a preparation of $|\overline{0}_{\sigma^2}^{\text{app}}\rangle$ as an $(s, \epsilon)$-independent Markovian noise model, where $\epsilon$ is expected to vary with the squeezing level $-10\log_{10}(2\sigma^2)$ for a fixed $s$.

Figure 5a shows a numerical estimate of $\epsilon$ for a fixed value $s = \sqrt{\pi}/14$ (see Supplementary Remark 38 for the reason to choose this value) with varying the squeezing level $-10\log_{10}(2\sigma^2)$, which is computed by comparing $|\overline{0}_{\sigma^2}^{\text{app}}\rangle$ with a renormalized version of $\hat{\Pi}_s|\overline{0}_{\sigma^2}^{\text{app}}\rangle$. As the figure suggests, our bound does not support the numerical thresholds of 10–20 dB in the literature[34,35,37,39,40,61,62], mainly because our analysis focuses on the theoretical rigor. However, the noise strength $\epsilon$ is below the fault-tolerant threshold of qubit concatenated codes[43] when the squeezing level is around 30 dB. There is also a caveat for this bound since $\hat{\Pi}_s|\overline{0}_{\sigma^2}^{\text{app}}\rangle$ has an infinite energy in general. See Supplementary Note 2C for a detailed discussion.

There are several typical noisy channels in CV systems. Among them, random displacement channels are easy to analyze with the GKP code since the GKP code is designed to correct small displacements. In fact, an energy constraint is not necessary to obtain an upper bound on

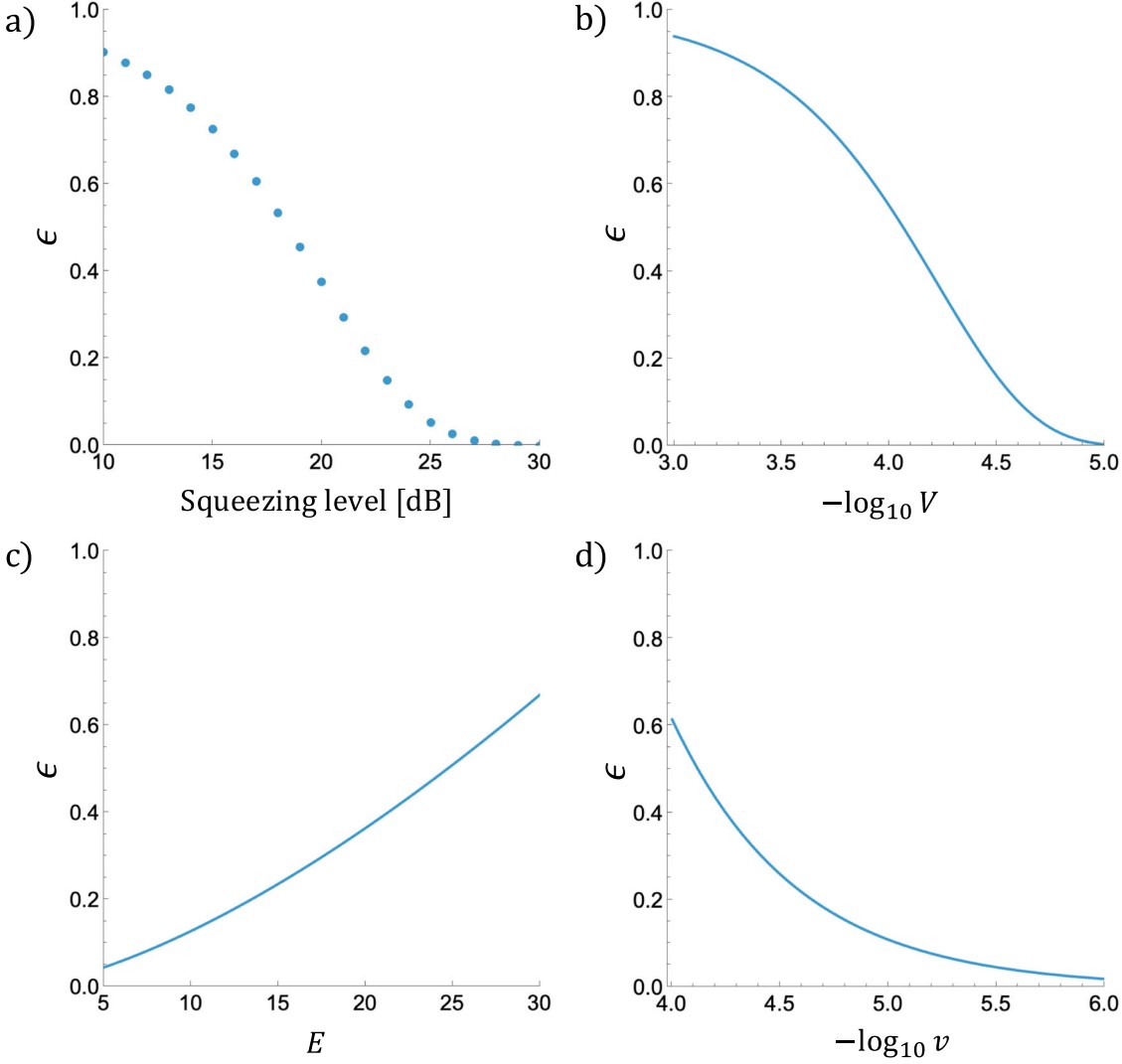

**Fig. 5 | Numerical estimates of the parameter $\epsilon$ of $(s, \epsilon)$- or $(E, s, \epsilon)$-independent Markovian noise models. a** shows the estimate of $\epsilon$ when a Gaussian-approximate GKP state $\left|\overline{0}_{\sigma^2}^{\text{app}}\right\rangle$ is regarded as a $(\sqrt{\pi}/14, \epsilon)$-independent Markovian noise model with varying the squeezing level $-10\log_{10}(2\sigma^2)$ in decibel. **b** shows the estimate of $\epsilon$ when a Gaussian-random displacement channel with the variance $V$ for both quadratures is regarded as an $(E, \sqrt{\pi}/182, \epsilon)$-independent Markovian noise model, which is independent of $E$. **c, d** shows the estimates when a random phase rotation channel with a wrapped normal distribution with the variance $v$ is regarded as $(E, \sqrt{\pi}/182, \epsilon)$-independent Markovian noise models, where **c** fixes the variance $v = 5 \times 10^{-5}$ and varies the energy $E$ while **d** fixes the energy $E = 20$ and varies the variance $v$.

$\epsilon$ when a random displacement channel is interpreted as $(E, s, \epsilon)$-independent Markovian noise to the identity channel. Figure 5b shows a numerical estimate of $\epsilon$ for a Gaussian random displacement noise with the variance $V$ in both quadratures for a fixed value of $s = \sqrt{\pi}/182$ (see Supplementary Remark 38 for the reason to choose this value), where the horizontal axis shows the minus logarithm of the variance $V$. The figure shows that even without an energy constraint, the noise strength $\epsilon$ can be small for a small variance $V$.

For conventional CV noise channels other than random displacements, one needs energy constraints to obtain a nontrivial bound on $\epsilon$. References [69,70] show some analytical upper bounds on energy-constrained diamond norms for optical loss channels and other experimentally relevant noise channels. They can directly be interpreted as $(E, 0, \epsilon)$-independent Markovian noise acting on the identity channel. See also numerical bounds therein. Random phase rotation channels are physically relevant yet not covered in their analyses, so we estimate $\epsilon$ for them when regarded as $(E, 0, \epsilon)$-independent Markovian noise to the identity channel. When a random phase is distributed according to a probability distribution $f(\theta)$ with $\theta \in [-\pi, \pi)$, then we have

$$\epsilon \leq \int_{-\pi}^{\pi} d\theta f(\theta) \sqrt[3]{4|\theta|E}. \tag{20}$$

See Supplementary Note 1D for the details. Figure 5c, d shows the behavior of $\epsilon$ when $f(\theta)$ is a wrapped normal distribution with the variance $v$, where 5c fixes the variance $v = 5 \times 10^{-5}$ and varies the energy $E$, while 5d fixes the energy $E = 20$ and varies the variance $v$. One can see that the requirement for the phase stability is quite high even for a moderate value of $E = 20$.

For measurements, an experimental homodyne detector has a finite resolution $b$ and a finite range $\Gamma$. When regarded as $(E, s, \epsilon)$-independent Markovian noise to the ideal homodyne measurement, we have $s = b$ and $\epsilon \leq (2E + 1)/\Gamma^2$. See Supplementary Note 1D for the detailed derivation.

In conclusion, the conventional noise models in CV systems can be reinterpreted as our newly defined $(s, \epsilon)$- or $(E, s, \epsilon)$-independent

Markovian noise. Thus, our result opens a way towards comprehensive understanding on error correctability in CV systems.

## Code availability
The code used in this work is available from the corresponding author upon request.

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

## Acknowledgements

T.M. acknowledges helpful discussions with Yui Kuramochi. T.M. was supported by JST CREST Grant Number JPMJCR23I3, JSPS Overseas Research Fellowships, JST PRESTO Grant Number JPMJPR24FA, and JST CREST Grant Number JPMJCR25I4. H.Y. was supported by JST PRESTO Grant Number JPMJPR201A, JPMJPR23FC, JSPS KAKENHI Grant Number JP23K19970, and JST CREST Grant Number JPMJCR25I5. This work was supported by the Australian Research Council (ARC) Center of Excellence for Quantum Computation and Communication Technology (Project No. CE170100012). N.C.M. was supported by an ARC Future Fellowship (Project No. FT230100571).

## Author contributions

T.M. contributed to the conception, analysis, and interpretation of the work and mainly wrote the manuscript. H.Y. contributed to the conception, analysis, and interpretation of the work and supervised the manuscript writing. N.C.M. contributed to the interpretation of the work and supervised the manuscript writing.

## Competing interests

The authors declare no competing interests.
