## [Transparent Peer Review File · Nature Communications]

Continuous-Variable Fault-Tolerant Quantum Computation under General Noise

Corresponding Author: Dr Takaya Matsuura

Version 0:

Reviewer comments:

Reviewer #1

(Remarks to the Author)

The manuscript by Matsuura, Menicucci and Yamasaki considered fault-tolerant quantum computation (FTQC) with the Gottesman-Kitaev-Preskill (GKP) code under Markovian-type noise. The current work generalizes the FTQC theories in two prior works, one studies the FTQC of GKP codes for specific Gaussian random displacement noise, and the FTQC of qubit quantum error correction code with general Markovian noise. Despite the extensive studies for the latter, it is nontrivial to generalize FTQC to the GKP code, primarily because ideal GKP code is unphysical and one needs to consider finite-energy approximated GKP code. For that, the authors adopted a stabilizer-subsystem decomposition approach to convert the physical noise in the GKP codes to qubit-level noise that are correctable via concatenated code. The authors generalized the gadgets used in the FTQC to constraint not only the spread of errors between different blocks but also the energy and displacements of the GKP codes, a key step to prove the level reduction theorem.

The manuscript is organized in a careful manner, with essential ideas summarized in the main text, and the full theory described comprehensively in the Supplementary Information. We believe that the manuscript may be publishable in Nature Communication if the authors address the comments below.

First, the advantage of using the stabilizer-subsystem decomposition to formulate the FTQC theory is unclear, particularly comparing to simpler methods of analyzing finite-energy GKP codes (e.g., based on twirling where a coherent Gaussian envelope is approximated as an incoherent Gaussian random displacement) which have been used in prior works, including Ref. 34-40 cited. It would make the impact of the work clearer if the authors can elaborate how their formalism can be used to obtain better results (e.g., more accurate error threshold estimation) compared to simpler alternatives.

Second, although the authors prove the existence of a threshold for FTQC with the GKP code, the required resource overheads for FTQC is not clearly presented. This is in contrast to the familiar threshold theorem for the qubit case where the number of gates needed for FTQC is estimated as a function of error rate and target accuracy. For example, in Theorem 1, a natural question is how many concatenations is needed to reach the target p given a set of (E, ϵ, r) . The authors seem to provide some calculations in Theorem S36 and Corollary S37 in the Supplement, but the result is not as succinct as the qubit threshold theorem [for example, Eq. 10.116 in Nielsen and Chuang], and it is natural to wonder if the result could be simplified further.

In a similar spirit as the previous remark, it may be useful to show a scaling of (E, ϵ, r) for given thresholds. For example, the author in Ref. 34 showed the squeezing required to achieve the thresholds for the Gaussian random displacement noise. Similar analysis for more general Markovian noise will be useful to guide the development of FTQC based on GKP codes.

Lastly, incorporation of "analog information" (or soft information) in decoding has been important for boosting the performance of concatenated GKP codes. It would be helpful if the authors can comment on whether their formalism can incorporate the analog information and more explicitly quantify the advantage of continuous-variable GKP codes over standard qubit codes.

Reviewer #2

(Remarks to the Author)

Reviewer #3

(Remarks to the Author)

Summary of paper

The authors provide a fault-tolerance threshold theorem for CV quantum computation based on GKP codes and concatenated qubit codes. Concretely, they define a class of noisy operations and corresponding metrics measuring the strength of the noise such that scalable universal computation is possible assuming the noise is sufficiently weak. Here it is assumed that the noise acts independently on each mode. The arguments establish the existence of a threshold error strength without giving quantitative bounds for it.

The main result builds on a seminal threshold theorem for qubit fault-tolerance under local Markovian noise. It is shown that when a GKP code is used to realize each qubit, the considered noise translates into local Markovian noise on the qubit level.

To formalize what type of noise can be accounted for, the authors rely on a stabilizer subsystem-(SSS)-decomposition of the Hilbert space of a mode. In principle, this captures e.g., inaccuracies introduced by the use of imperfect (non-ideal) GKP states, as well as noise channels whose Kraus operators are bounded displacements. In addition, the results are generalized to noise that is close in the energy-constrained diamond norm to such channels.

Discussion

Establishing a fault-tolerance threshold theorem for CV systems is challenging partly because identifying a suitable set of noise models is non-trivial. For example, continuity bounds (e.g., based on the diamond norm) are not immediately applicable (as in the finite-dimensional case). In prior work, fault-tolerance thresholds were only established with respect to displacement noise, which is analogous to the restricted model of random Pauli errors in the qubit stabilizer setting.

This submission works towards generalizing this constraint, claiming to establish a fault-tolerance threshold theorem against general CV noise. Indeed, the SSS-decomposition of the Hilbert space provides an elegant way of reasoning e.g., about about imperfect GKP states (one source of error when using this kind of encoding). In addition, the use of energy-constrained diamond norms is shown to allow to capture noise channels that are close (in this norm). I do have some concerns/questions about the degree of generality, as well as the energy constraints imposed here, however (see below).

The fault-tolerance approach, i.e., the reduction to local Markovian qubit noise is certainly natural and leverages seminal existing work in this direction.

Concerning the noise model, I have the following questions:

- First, given that the SSS-decomposition allows to discuss general (imperfect) GKP states, it would be good to have concrete examples. This concerns e.g., finitely squeezed GKP states with a Gaussian envelope. To have any experimental relevance, it would be necessary to understand the relationship between the degree of squeezing and e.g., the parameter r (i.e., the size of the square region where the state is supported). I am not sure why there is not even a single example of this kind – is there some fundamental obstacle to making this translation?
- While the general setup and discussion is supposed to capture “general noise”, the definition of an s -parametrized noise channel (central to the analysis) uses only displacements of bounded size (i.e., the Kraus operators need to be linear combinations of such operators). It is not clear to me how limiting this constraint is, and, more importantly, whether it is even necessary. It appears to me that a more general definition could be considered (which basically discusses how the square region gets deformed under noise).

Once again, concrete examples would be useful to see the degree of generality/how these parameters translate into practically relevant examples. For example, for a Gaussian single-mode attenuation channel, or a single-mode phase space rotation, it would be good to know what the parameter s looks like.

With regards to the energy-bounded norms, the authors argue and use that the energy stays bounded during the computation. This is clear for the unitary operations (according to standard arguments), but the authors also argue that this is the case for the Knill-type GKP error correction used. These arguments are not sufficiently detailed, in my opinion and should be substantiated in the main text. Specifically, it appears that the energy (also of modes that are not being measured) could increase by measurement – although this may not be the case on average, it will certainly happen with some probability. It is unclear to me why the average energy should be relevant here, as the arguments should apply to any post-measurement state. After all, these states are subsequently used in the computation.

Recommendation

The paper is generally well written and relatively easy to follow apart from some of the points mentioned below. The contribution is certainly interesting. At the same time, it seems that central technical issues should be clarified (see above). Furthermore, the lack of concrete examples (relating noise thresholds to physically relevant quantities) is a negative point. In summary, although I believe this paper makes an interesting contribution, it requires more work in order to become publishable, in my opinion.

A few editorial comments/typos

The manuscript is littered with typos and should be proofread by a native English speaker. Most unpleasant for the reader is the systematic incorrect use of the article “the”, starting with the abstract:

The quantum error-correcting code [...] attracts much attention due to its flexibility...
It is unclear what “THE” code is (there are several CV error-correcting codes), what is meant is probably “Quantum error-correction with CV systems attracts...”

Similarly,
“we show that THE Markovian-type noise in CV systems is translated into THE Markovian-type noise logical qubits”.
Having shown “the upper bound”

I could list numerous (>15) occurrences of such incorrect uses of articles (for example, the authors do not establish “THE threshold theorem for CV quantum computation” but “A threshold theorem for CV quantum computation”, “the s-parametrized noise channel”)

Given the poor level of proof-reading, I will not list other typos here at this point.

p. 5: The way the threshold theorem is stated here seems quite incomplete. In particular, for fault-tolerance considerations, it is crucial to understand what overhead is involved. This should be part of the statement of the theorem.

p. 8: The argument for level reduction is relatively clear, up to the fact that it should probably also be argued that local Markovian noise remains local Markovian when going up a level in concatenation.

p. 10: The definition of the GKP *-decoder is unclear, it should be given explicitly as a CPTP map (is there some tracing out involved?) Symbols U , $\text{cal}U$ in Eq. 16 should be explained. Similarly, Definition 2 is unclear, perhaps a diagrammatic example could help to clarify this.

p. 10: Would it be possible to formulate a FT-threshold theorem at this point, i.e., for s-parametrized noise channels (without taking into account energy considerations)? Perhaps this could clarify the presentation.

p. 11: Hermitian-preserving instead of Hermetian-preserving (at least 5 times).
References should be given for the first four statements, which supposedly are not new.

p. 15: The delta-distribution “on the interval” $(-a/2, a/2)$ should be properly defined in formulas instead of stating something unclear in words.

Version 1:

Reviewer comments:

Reviewer #1

(Remarks to the Author)

We have read the authors' responses to the referees as well as the updated manuscript. The authors have addressed all the questions, and edited the manuscript based on the feedback. Hence, we recommend to accept the manuscript for publication.

Reviewer #2

(Remarks to the Author)

Reviewer #3

(Remarks to the Author)

The authors have made minor changes to the manuscript which address some of my comments. Unfortunately, I cannot say that my main criticism has been addressed in a convincing manner. This concerns especially the lack of explicit examples.

For Gaussian-envelope approximations to GKP code words, the authors added a comment referring to Appendix A4. The only addition in that appendix is a single sentence explaining the standard notion of squeezing (the way squeezing is measured), and a comment saying that the list of examples is incomplete. There are still no explicit bounds/relationships between e.g., squeezing parameter and (r, ϵ) as would be desirable. For general (even simple Gaussian) noise channels, there are no non-trivial examples with parameter $s \neq 0$ - this is only given for homodyne detection. In particular, there are no additional substantial remarks about this point in the main paper.

In their rebuttal letter, the authors argue that “obtaining a precise estimate of the threshold value requires massive numerics since the minimization should be taken over all possible functions that satisfy certain constraints. Obtaining a threshold value numerically is not the aim of this work [...]” It is certainly the case that obtaining precise estimates will be challenging (in fact, it is not clear how to perform this kind of optimization at all). However, this is not what I was asking for (nor, I believe the other reviewers): It would already be satisfactory to have some (even conservative) bounds, for example.

If it is not possible to give analytic trade-offs, I suggest moderating the claims made in the paper. For example, the abstracts talks about “an explicit bound on the noise strength”, which, as the authors argue in their rebuttal letter, is in fact not given: all that is shown is the existence of a threshold.

While I still consider the framework interesting, the lack of explicit bounds makes this somewhat less practically relevant. This is perhaps ok (as this is a theory contribution), but claiming e.g., that this closes “the aforementioned gap between [...] FT theory and current as well as future experiments of CV quantum computing” then appears to be a bit of an exaggeration.) If the main point is the existence of a threshold, this should be emphasized and/or the suggested numerical difficulty/impossibility of getting actual threshold values should be mentioned in the main paper.

(It is standard practice to compute/estimate threshold values in e.g., the qubit context, and in the past, this has been done even in cases where the corresponding values were not necessarily experimentally realistic.)

In summary, my main assessment remains unchanged: in its current form, I'm not fully convinced of this contribution because the applicability of this framework (in order to obtain explicit bounds) is not demonstrated.

Version 2:

Reviewer comments:

Reviewer #3

(Remarks to the Author)

The authors have added some numerical data and discussion to solidify the idea that this framework can provide concrete numerical threshold estimates. While I do not find all these arguments entirely convincing (see below), they do add to the paper, and may lead to follow-up work with more explicit estimates. I am also happy to see that some of the claims have now been moderated (but see the remarks below).

In more detail:

- The newly introduced numerical values in Fig. 5 and the associated discussion for states is derived using an infinite-energy-state, and it is a priori unclear if such bounds have much meaning if translated to more realistic states. I appreciate that the authors mention this caveat prominently, at least. (The discussion in the supplementary material does not fully work out the desired tradeoff, see p. 37.)

- I understand the point (now explained in more detail) that a bound with $s=0$ for noise channels trivially implies one for $s \neq 0$. But this does not make full use of the introduced framework (which presumably could provide better bounds), which is somewhat unfortunate.

- I appreciate the elimination of the term “explicit bound” in places where no explicit bound was known or provided (following the established terminology that an explicit bound is an analytical formula depending on the given parameters). However, I do not fully understand what is meant by the newly introduced phrasing

“with clarification of an upper bound on the noise strength”

(used in the abstract and also later in the paper). It is quite unclear, in my opinion, what “clarification” means in such a phrase – this is not standard terminology. Perhaps this can be omitted altogether?

One additional comment: To clarify the meaning of the GKP *-decoder etc, it would be good to include a reference to <https://arxiv.org/abs/0904.2557> where the meaning of such a decoder and the need is discussed in detail.

Reply to the reviewers' comments

In the following, the reviewers' comments are quoted in blue text. Changes made are highlighted with red text.

Reviewers 1 & 2 (Co-review)

The manuscript by Matsuura, Menicucci and Yamasaki considered fault-tolerant quantum computation (FTQC) with the Gottesman-Kitaev-Preskill (GKP) code under Markovian-type noise. The current work generalizes the FTQC theories in two prior works, one studies the FTQC of GKP codes for specific Gaussian random displacement noise, and the FTQC of qubit quantum error correction code with general Markovian noise. Despite the extensive studies for the latter, it is non-trivial to generalize FTQC to the GKP code, primarily because ideal GKP code is unphysical and one needs to consider finite energy approximated GKP code. For that, the authors adopted a stabilizer-subsystem decomposition approach to convert the physical noise in the GKP codes to qubit-level noise that are correctable via concatenated code. The authors generalized the gadgets used in the FTQC to constraint not only the spread of errors between different blocks but also the energy and displacements of the GKP codes, a key step to prove the level reduction theorem. The manuscript is organized in a careful manner, with essential ideas summarized in the main text, and the full theory described comprehensively in the Supplementary Information. We believe that the manuscript may be publishable in *Nature Communication* if the authors address the comments below.

We appreciate the reviewers' careful reading of our manuscript and a concise summary of our manuscript. Since we have addressed the reviewers' comments, we believe our revised manuscript is publishable in *Nature Communications*.

First, the advantage of using the stabilizer-subsystem decomposition to formulate the FTQC theory is unclear, particularly comparing to simpler methods of analyzing finite-energy GKP codes (e.g., based on twirling where a coherent Gaussian envelope is approximated as an incoherent Gaussian random displacement) which have been used in prior works, including Ref. 34-40 cited. It would make the impact of the work clearer if the authors can elaborate how their formalism can be used to obtain better results (e.g., more accurate error threshold estimation) compared to simpler alternatives.

We thank that the reviewers raised a question on the comparison to previous works. The twirling-like method developed in Ref. [37] in the previous and revised manuscript conceptually works for converting coherent superposition of displacement noise to random displacement noise, which thus works for analyzing the ultimate limit of the error-correcting performance of the GKP code. However, this twirling-like operation itself cannot be physically realizable since it requires an infinite amount of energy to carry out. Furthermore, a fault-tolerant analysis requires how a noise model behaves during a computation with a fixed error-correcting procedure, inserting an auxiliary channel to simplify the problem is prohibited. For example,

a quantum error-correcting code may have a worse threshold against over-rotation noise than against its twirled version, which is stochastic Pauli noise, even though the channel capacity of the former noise is no smaller than that of the latter. To emphasize these points, we add the following text to Introduction.

Reference [37] proposed a twirling-like method that reduces a Gaussian-approximate GKP state to an ideal GKP state subjected to Gaussian random displacement noise, which may be applicable to other types of noise. However, this twirling-like operation cannot be physically realizable. Furthermore, virtually inserting a channel during the computation changes noise models, and therefore, this simplification cannot be used for a fault-tolerance analysis.

Second, although the authors prove the existence of a threshold for FTQC with the GKP code, the required resource overheads for FTQC is not clearly presented. This is in contrast to the familiar threshold theorem for the qubit case where the number of gates needed for FTQC is estimated as a function of error rate and target accuracy. For example, in Theorem 1, a natural question is how many concatenations is needed to reach the target p given a set of (E, ϵ, r) . The authors seem to provide some calculations in Theorem S36 and Corollary S37 in the Supplement, but the result is not as succinct as the qubit threshold theorem [for example, Eq. 10.116 in Nielsen and Chuang], and it is natural to wonder if the result could be simplified further.

The goal of our result is to map a noise model as well as a noise parameter at the continuous-variable (CV) physical level to those at the logical-qubit level of the GKP code. Once this is achieved, we can apply the well-established theory of fault-tolerant quantum computation with a qubit concatenated code. Thus, the resource overheads for FTQC are determined by those of a qubit concatenated code as long as the translated noise parameter from the CV level is below the threshold of a qubit concatenated code. In this sense, the only difference from the qubit case is that the physical noise strength at the bottom layer of concatenation is replaced with the noise strength that is translated from the CV-level noise. Putting aside these, we think an explicit remark on the resource overhead is helpful for readers' understanding, so we added the following paragraph at the end of Method.

In particular, Ref. [?] shows that, by concatenating the Steane code, the noise strength at the k^{th} level of concatenation decreases doubly exponentially as $\mathcal{O}(c^{2^k})$ with $c := \epsilon_{\text{qubit}}/\epsilon_{\text{qubit}}^* < 1$ while the space and time overhead increases exponentially in k . Thus, a constant accuracy ϵ for a logical outcome of W -qubit D -depth circuit can be achieved within $\mathcal{O}(\text{poly}(\log_{1/c}(WD/\epsilon)))$ space and time overheads [32].

We also add the same statement in the corresponding corollary (Corollary S37) in Supplementary Information.

In a similar spirit as the previous remark, it may be useful to show a scaling of (E, ϵ, r) for given thresholds. For example, the author in Ref. 34 showed the squeezing required to achieve the thresholds for the Gaussian random displacement noise. Similar analysis for more general Markovian noise will be useful to guide the development of FTQC based on GKP codes.

We thank the referees for highlighting this point. Indeed, a scaling of (E, ϵ, r) for a given threshold is calculable for a fixed noise model. We actually give some examples of such scaling in Appendix A4 for several noise models that are frequently considered in quantum optics, but they were not the full list of relevant noise models nor the full analysis of scaling behaviors between these parameters. A full scaling analysis should be carried out with numerical simulations, but here our aim is to derive an analytic bound on the fault-tolerant threshold. Thus, we leave the problem to future work and instead add the following remark at the end of Appendix A4 in the revised manuscript.

These are not the full list of relevant noise models in quantum optical setups nor the full tradeoff analysis of parameters for a given noise model, while our contribution is to find a class of noise models against which FT threshold exists. Deriving general scaling behaviors between parameters (E, s, ϵ) is left open to future work.

Lastly, incorporation of “analog information” (or soft information) in decoding has been important for boosting the performance of concatenated GKP codes. It would be helpful if the authors can comment on whether their formalism can incorporate the analog information and more explicitly quantify the advantage of continuous-variable GKP codes over standard qubit codes.

We appreciate your pointing it out since the incorporation of analog information in a decoder often dramatically changes numerical thresholds for Gaussian random displacement noise. In our framework, analog information will give nontrivial information to a decoder as long as one can assume the shape of a wave function on the syndrome subsystem. Existing analyses amount to assuming a Gaussian-like wave function on the syndrome subsystem, which may be reasonable as long as noise in the circuit is only Gaussian random displacement. For more general noise, the shape of the wave function may not be Gaussian-like, and thus analog outcomes may not have useful information for a decoder. In any case, utilizing analog information in a decoder is a noise-dependent heuristics that may improve the performance only empirically without theoretical guarantee; it may not imply a rigorous bound on the threshold as we aim to do in this work. To explain these points, we add the following sentence to the Discussion section in the revised manuscript.

We comment on how the use of analog information in the decoding affects our analysis, since the use of CV measurement outcomes often improves the threshold dramatically [35–37, 39, 40, 62, 63]. It is possible to gain nontrivial information from CV measurement outcomes only if some assumptions on the shape of a wave function on \mathcal{H}_S is imposed. Existing analyses assume a Gaussian-like wave function on \mathcal{H}_S , which may no longer be justified for a general type of noise in our analysis. Thus, the use of analog information should be regarded as a noise-dependent heuristic rather than a general strategy to obtain a rigorous threshold, which we aim to achieve in this paper.

Reviewer 3

Establishing a fault-tolerance threshold theorem for CV systems is challenging partly because identifying a suitable set of noise models is non-trivial. For example, continuity bounds (e.g., based on the diamond norm) are not immediately

applicable (as in the finite-dimensional case). In prior work, fault-tolerance thresholds were only established with respect to displacement noise, which is analogous to the restricted model of random Pauli errors in the qubit stabilizer setting.

This submission works towards generalizing this constraint, claiming to establish a fault-tolerance threshold theorem against general CV noise. Indeed, the SSS-decomposition of the Hilbert space provides an elegant way of reasoning e.g., about imperfect GKP states (one source of error when using this kind of encoding). In addition, the use of energy-constrained diamond norms is shown to allow to capture noise channels that are close (in this norm). I do have some concerns/questions about the degree of generality, as well as the energy constraints imposed here, however (see below).

The fault-tolerance approach, i.e., the reduction to local Markovian qubit noise is certainly natural and leverages seminal existing work in this direction.

We appreciate the reviewer’s evaluation on why our work is important and how the existing problem is overcome by our method. We also thank the reviewer for the critical review of the details of our results, since it helps us to improve our paper.

Concerning the noise model, I have the following questions:

- First, given that the SSS-decomposition allows to discuss general (imperfect) GKP states, it would be good to have concrete examples. This concerns e.g., finitely squeezed GKP states with a Gaussian envelope. To have any experimental relevance, it would be necessary to understand the relationship between the degree of squeezing and e.g., the parameter r (i.e., the size of the square region where the state is supported). I am not sure why there is not even a single example of this kind — is there some fundamental obstacle to making this translation?

We thank the referee for raising this point, as the connection between our noise parameters and the commonly used squeezing parameter was not clearly explained in the previous version of the manuscript. We discuss how the conventional approximate GKP state can be represented in terms of the stabilizer subsystem decomposition in Appendix A4 in the previous and revised manuscript. (Appendix A4 also contains other physically relevant examples such as the attenuation channel and the random phase rotation channel, which we will come back to later.) In fact, there is a tradeoff between the preparation energy constraint E_{prep} , which we can choose, and the parameters (r, ϵ) of the (r, ϵ) -independent Markovian noise. Thus, one can draw a tradeoff curve between these parameters in principle. However, obtaining a precise estimate of the threshold value requires massive numerics since the minimization should be taken over all the possible functions that satisfy certain constraints. Obtaining a threshold value numerically is not the aim of this work, but we solve a more fundamental problem regarding its existence. We thus leave this evaluation and evaluation of similar tradeoff relations for other noise models to future work. Making examples we considered more explicit, we add the following sentence to the Method section.

Explicit examples of the noise model, such as the conventional Gaussian approximation of the GKP codeword [9,42], random phase rotation, and a finite resolution of homodyne measurement, are also given in Sec. A4 of Supplementary information

We further add the following sentence to Appendix A4 in the revised manuscript to make the connection to the conventional evaluation of approximation (i.e., squeezing level) more explicit.

The ratio in decibel between the variance of each peak of the position probability distribution of $|\bar{0}_{\sigma^2}^{\text{app}}\rangle$ and that of the vacuum, i.e., $-10 \log(2\sigma^2)$, is called the squeezing parameter of this approximate GKP state.

We hope these changes make readers easier to understand the relationship between the conventional parameterization and our new parameterization.

- While the general setup and discussion is supposed to capture “general noise”, the definition of an s -parameterized noise channel (central to the analysis) uses only displacements of bounded size (i.e., the Kraus operators need to be linear combinations of such operators). It is not clear to me how limiting this constraint is, and, more importantly, whether it is even necessary. It appears to me that a more general definition could be considered (which basically discusses how the square region gets deformed under noise).

Once again, concrete examples would be useful to see the degree of generality/how these parameters translate into practically relevant examples. For example, for a Gaussian single-mode attenuation channel, or a single-mode phase space rotation, it would be good to know what the parameter s looks like.

The question raised by the reviewer here is split into two questions: how limiting an s -parameterized noise channel is, and why authors limit their attention to displacements of bounded size. For the first question, our s -parameterized noise channel is a very limited class of noise models since physically relevant noise models that are exactly written as an s -parameterized noise channel are only deterministic and random displacement channels with a small amount. We would like to stress, though, that the role of s -parameterized noise channel here is to introduce a perfectly correctable noise, as the weight- t Pauli noise with t smaller than half of the distance is perfectly correctable by a qubit $[[n, k, d]]$ code. For general noise models, we consider approximating them with an s -parameterized noise channel, as general Markovian noise models are decomposed into Pauli operators in a qubit code.

For the second question, more general deformations of the square region could be considered, but such deformations do not satisfy the fault-tolerance condition in general. Here, an s -parameterized noise channels are carefully defined so that ideal gates and measurements do not amplify the parameter s . More general deformations need not be preserved under ideal gate operations. In this sense, gates and measurements listed in our paper, which are also conventionally used in literature, satisfy similar properties to “transversal operations” in a qubit code, as explained at the beginning of page 10 of the revised manuscript.

There is thus a tradeoff between the parameter s and ϵ in (E, s, ϵ) -independent Markovian noise. For the attenuation channel and the phase rotation channel, we currently have a bound on ϵ for a fixed E and $s = 0$, as explained in Appendix A4 in the revised manuscript. Again, we leave more general tradeoff relations to future work.

The confusion may come from that our explanations on how to handle general noise models in the Method section were not well-organized. Thus, we add the following sentence before starting the explanation of the fault-tolerant gadgets.

In the following, we first define classes of preparation, gate, measurement, and EC gadgets with parameters representing spread of displacement errors in the gadgets. We then introduce fault-tolerance conditions for these gadgets, which essentially require that errors in the gadgets do not spread too much and the gadgets implement their intended logical operations if errors are correctable. We call the gadgets satisfying these conditions fault-tolerant gadgets. The operations in Eqs. (1)–(4) satisfy these conditions. Finally, we consider approximating general noise models with these fault-tolerant gadgets, which will be discussed in the next section.

With regards to the energy-bounded norms, the authors argue and use that the energy stays bounded during the computation. This is clear for the unitary operations (according to standard arguments), but the authors also argue that this is the case for the Knill-type GKP error correction used. These arguments are not sufficiently detailed, in my opinion and should be substantiated in the main text. Specifically, it appears that the energy (also of modes that are not being measured) could increase by measurement – although this may not be the case on average, it will certainly happen with some probability. It is unclear to me why the average energy should be relevant here, as the arguments should apply to any post-measurement state. After all, these states are subsequently used in the computation.

We apologize for not explicitly explaining why measurement-adaptive operations satisfy energy-constraint conditions in the previous manuscript. The reason we only need to consider the average case comes from the definition of the fault tolerance; the output probability distribution of the fault-tolerant circuit should be close to that of the ideal circuit when irrelevant outcomes, such as those used in the error corrections, are traced out. In general, the output probability distribution of an FT circuit when conditioned on particular EC measurement outcomes may be far from that of an ideal circuit, even in a qubit FTQC. Thus, even though there may be a case in which the energy-constraint conditions are violated with a rare measurement outcome during an execution of CV quantum computation, it is not a problem as long as its probability is small. The reviewer’s question is natural since previously we wrote as follows in the main text:

For this, the mode-wise energy (i.e., the average photon number) of the state at every time step during the computation needs to be bounded.

It is unclear from this whether “the mode-wise energy of a state” means that of an averaged state or a conditional state. We thus change it as follows:

For this, the mode-wise energy (i.e., the average photon number) of a state at every time step during computation needs to be bounded when averaged over measurement outcomes.

Furthermore, we add the following short sentence to the beginning of page 8:

... the circuit C' outputs a W -bit string sampled from a probability distribution close to the output probability distribution of the original circuit C within error in the total variation distance at most ϵ when irrelevant measurement outcomes, such as those that appear in EC steps, are traced out [33].

Finally, we add the following sentences to page 12 of the Method section:

Note that we imposed the requirement that the energy of a state averaged over EC measurement outcomes should be bounded. This is because our ultimate goal is to show that the output probability distribution of this FT-GKP circuit C' is close to that of the qubit circuit C , when measurement outcomes in EC steps are traced out.

Recommendation

The paper is generally well written and relatively easy to follow apart from some of the points mentioned below. The contribution is certainly interesting. At the same time, it seems that central technical issues should be clarified (see above). Furthermore, the lack of concrete examples (relating noise thresholds to physically relevant quantities) is a negative point.

In summary, although I believe this paper makes an interesting contribution, it requires more work in order to become publishable, in my opinion.

We first thank the reviewer for evaluating our work as an interesting contribution. We expect that the technical concerns pointed out by the reviewer are well-addressed by our revised manuscript as explained above. We think that the reviewer may have overlooked the concrete examples in Appendix A4, which may be the reason why the reviewer pointed out the lack of concrete examples. In the main text of the revised manuscript, we added an instruction to the examples in the appendix so that readers can find them more easily. With the following revisions to typos and unclear terminology, we hope these changes make the contribution and the technical correctness of our work clearer enough for publication.

The manuscript is littered with typos and should be proofread by a native English speaker. Most unpleasant for the reader is the systematic incorrect use of the article “the”, starting with the abstract:

The quantum error-correcting code [...] attracts much attention due to its flexibility...

It is unclear what “THE” code is (there are several CV error-correcting codes), what is meant is probably “Quantum error-correction with CV systems attracts....”

Similarly,

“we show that THE Markovian-type noise in CV systems is translated into THE Markovian-type noise logical qubits”. Having shown “the upper bound”

I could list numerous (15) occurrences of such incorrect uses of articles (for example, the authors do not establish “THE threshold theorem for CV quantum computation” but “A threshold theorem for CV quantum computation”, “the s -parametrized noise channel”) Given the poor level of proof-reading, I will not list other typos here at this point.

Thank you for the comment. In our revised version, we used a grammar checker.

p. 5: The way the threshold theorem is stated here seems quite incomplete. In particular, for fault tolerance considerations, it is crucial to understand what overhead is involved. This should be part of the statement of the theorem.

We changed the last statement of Theorem 1 as follows:

That is, by concatenating this CV protocol with a fault-tolerant protocol for a qubit concatenated code with a threshold $p_{\text{th}} > p$ under local Markovian noise, we can achieve fault-tolerant quantum computation using CV systems at the physical level with an overhead polylogarithmic in the size of an original circuit.

We also add the explanation of this overhead estimation at the end of the Method section:

In particular, Ref. [32] shows that, by concatenating the Steane code, the noise strength at the k^{th} level of concatenation decreases doubly exponentially as $\mathcal{O}(c^{2^k})$ with $c := \epsilon_{\text{qubit}}/\epsilon_{\text{qubit}}^* < 1$ while the space and time overhead increases exponentially in k . Thus, a constant accuracy ϵ for a logical outcome of W -qubit D -depth circuit can be achieved within $\mathcal{O}(\text{poly}(\log_{1/c}(WD/\epsilon)))$ space and time overheads [32].

We also add the same statement in the corresponding corollary (Corollary S37) in Supplementary Information.

p. 8: The argument for level reduction is relatively clear, up to the fact that it should probably also be argued that local Markovian noise remains local Markovian when going up a level in concatenation.

Thank you for the comment. There has already been this explanation just after the explanation of a local Markovian noise model. For readability, we repeat this statement in the Supplementary information as follows.

In Ref. [32], it is shown that local Markovian noise at the k^{th} level of concatenation is translated to local Markovian noise at the $(k + 1)^{\text{th}}$ level of concatenation with a different noise strength in a qubit concatenated code.

p. 10: The definition of the GKP *-decoder is unclear, it should be given explicitly as a CPTP map (is there some tracing out involved?) Symbols U, \mathcal{U} in Eq. 16 should be explained. Similarly, Definition 2 is unclear, perhaps a diagrammatic example could help to clarify this.

We clarify that the GKP *-decoder is a unitary operation in the revised version. It simply changes the basis to the SSS Hilbert space. We added an explanation of \mathcal{U} as follows:

Here, \mathcal{U}' denotes a channel that acts jointly on the logical qubit and the syndrome subsystem. Finally, Definition 2 is just a repeated use of Eq. (16). We added an explanation in Definition 2 in the revised manuscript.

p. 10: Would it be possible to formulate a FT-threshold theorem at this point, i.e., for sparametrized noise channels (without taking into account energy considerations)? Perhaps this could clarify the presentation.

We appreciate the reviewer’s suggestion. We added the following statement to the beginning of page 11 for clarity.

Combined with the fact that each GKP ExRec has a constant number of locations, there exists a threshold value s_{th} such that if the CV FT-GKP circuit C' consists only of s_{p} -preparations, s_{g} -gates, and s_{m} -measurements with $s_{\text{p}}, s_{\text{g}}, s_{\text{m}} < s_{\text{th}}$, then all the GKP ExRecs in C' are good, implying that $\tilde{C} = C$.

p. 11: Hermitian-preserving instead of Hermetian-preserving (at least 5 times).
References should be given for the first four statements, which supposedly are not new.

Thank you for pointing out the misspelling. We corrected them. We further added appropriate references for the first three statements. The fourth statement is a new statement that we prove in Sec. A3 of the revised manuscript.

p. 15: The delta-distribution “on the interval” $(-a/2, a/2)$ should be properly defined in formulas instead of stating something unclear in words.

Thank you for pointing out the ambiguity of the definition. We avoid the use of the delta distribution on the interval and use the ordinary Dirac delta function instead by imposing restrictions on the variables as follows:

The Zak basis satisfies the following orthogonal property for $z_1, z'_1 \in [-\frac{c}{2}, \frac{3c}{2})$ and $z_2, z'_2 \in [-\frac{c}{2}, \frac{c}{2})$:

[Equation A20],

where $\delta(x)$ denotes the Dirac delta function.

Reply to the reviewers' comments

We thank the reviewers for their evaluations and constructive suggestions. The reviewers' comments are quoted in blue text below. Changes made in the second round of revision are highlighted in magenta text, while the changes in the first round are colored red.

Reviewers 1 & 2 (Co-review)

We have read the authors' responses to the referees as well as the updated manuscript. The authors have addressed all the questions, and edited the manuscript based on the feedback. Hence, we recommend to accept the manuscript for publication.

We thank the reviewers for their previous constructive feedback and for recommending our revised manuscript for publication.

Reviewers 3

The authors have made minor changes to the manuscript which address some of my comments. Unfortunately, I cannot say that my main criticism has been addressed in a convincing manner. This concerns especially the lack of explicit examples.

We are sorry for not clearly presenting our results, especially when connecting our noise model to conventional noise models. We have largely revised our manuscript to meet the reviewer's request. Please refer to the point-by-point response below.

For Gaussian-envelope approximations to GKP code words, the authors added a comment referring to Appendix A4. The only addition in that appendix is a single sentence explaining the standard notion of squeezing (the way squeezing is measured), and a comment saying that the list of examples is incomplete. There are still no explicit bounds/relationships between e.g., squeezing parameter and (r, ϵ) as would be desirable. For general (even simple Gaussian) noise channels, there are no non-trivial examples with parameter $s \neq 0$ - this is only given for homodyne detection. In particular, there are no additional substantial remarks about this point in the main paper.

In their rebuttal letter, the authors argue that "obtaining a precise estimate of the threshold value requires massive numerics since the minimization should be taken over all possible functions that satisfy certain constraints. Obtaining a threshold value numerically is not the aim of this work [...]" It is certainly the case that obtaining precise estimates will be challenging (in fact, it is not clear how to perform this kind of optimization at all). However, this is not what I was asking for (nor, I believe the other reviewers): It would already be satisfactory to have some (even conservative) bounds, for example.

For the Gaussian-envelope approximate GKP state, we have added a numerical study in the Methods section of the main text to provide an upper bound on the noise strength ϵ of the

(s, ϵ) -independent Markovian noise with a fixed value of s and varying the squeezing level of the Gaussian-approximate GKP state. This is calculated through the Zak representation of the Gaussian-approximate GKP state as given in Eq. (A105) in the Supplementary Information. There is, however, a caveat for this, as it calculates the fidelity to an infinite-energy s -parameterized GKP state. We have explicitly noted this caveat in the main text of the revised manuscript and outlined a possible route to address it in Sec. A3 of the Supplementary Information. For this purpose, the explanations in Sec. A3 have also been largely extended. For conventional CV noise channels, we have also added numerical bounds on the noise strength ϵ of the (E, s, ϵ) -independent Markovian noise. The reviewer might think that our examples are only for $s = 0$. However, an upper bound on ϵ of an $(E, 0, \epsilon)$ -independent Markovian noise is also an upper bound on that of (E, s, ϵ) with $s > 0$. In other words, our numerical bound simply serves as a looser lower bound for $s > 0$ at the same time. Furthermore, we have added an example of a random displacement channel and a numerical bound on the parameter ϵ of (E, s, ϵ) for it. In this case, an upper bound on ϵ nontrivially depends on s but not on E .

If it is not possible to give analytic trade-offs, I suggest moderating the claims made in the paper. For example, the abstracts talks about “an explicit bound on the noise strength”, which, as the authors argue in their rebuttal letter, is in fact not given: all that it is shown is the existence of a threshold.

The meaning of “an explicit bound on the noise strength” here meant that the noise strength of the local Markovian noise model for qubits made up of the GKP codes can be bounded by a function of ϵ of our (E, s, ϵ) -independent Markovian noise model at the CV level. Since it was not clearly written in the abstract, which also confused the reviewer, we have revised the corresponding sentence as follows.

..., with clarification of an upper bound on the noise strength in terms of our newly introduced noise parameterization.

Moreover, we have changed the word “explicit bound” throughout the manuscript to avoid confusion.

While I still consider the framework interesting, the lack of explicit bounds makes this somewhat less practically relevant. This is perhaps ok (as this is a theory contribution), but claiming e.g., that this closes “the aforementioned gap between [...] FT theory and current as well as future experiments of CV quantum computing” then appears to be a bit of an exaggeration.) If the main point is the existence of a threshold, this should be emphasized and/or the suggested numerical difficulty/impossibility of getting actual threshold values should be mentioned in the main paper.

(It is standard practice to compute/estimate threshold values in e.g., the qubit context, and in the past, this has been done even in cases where the corresponding values were not necessarily experimentally realistic.)

As the reviewer recognized, our focus is to show the existence of a fault-tolerant threshold against general noise in CV systems, which are presented as the theorem statements using

our newly introduced noise parameterization. However, we have added new materials to the Methods in our revised manuscript to show that, using our theory, one may obtain a fault-tolerant threshold for a conventional CV noise parameter as well. Still, we agree with the reviewer in that the statement “closing the aforementioned gap between [...] FT theory and current as well as future experiments of CV quantum computing” was potentially confusing, so we changed it as follows.

..., presenting a pathway to bridge the aforementioned gap between the existing fault-tolerance theory and current as well as future experiments of CV quantum computing.

In summary, my main assessment remains unchanged: in its current form, I’m not fully convinced of this contribution because the applicability of this framework (in order to obtain explicit bounds) is not demonstrated.

With a numerical upper bound on the noise strength added to the Methods section and the corresponding explanations in the Supplementary Information, we believe that the wide applicability of our results becomes clearer. As the reviewer recognized, obtaining a tight evaluation of the noise strength for each conventional CV noise model can be reasonably left for future study. However, we have demonstrated the existence of a fault-tolerant threshold. The materials added to the revised manuscript shows how to translate our results to conventional parameters of CV noise models. We thus believe that our revised manuscript is publishable from *Nature Communications*.

Reply to the reviewer's comments

We again thank the reviewer for the comments, which have been improving our manuscript. We have revised our manuscript reflecting the reviewer's comments with new changes given in red text. The one-to-one response to the reviewer's comments is given in the following.

The authors have added some numerical data and discussion to solidify the idea that this framework can provide concrete numerical threshold estimates. While I do not find all these arguments entirely convincing (see below), they do add to the paper, and may lead to follow-up work with more explicit estimates. I am also happy to see that some of the claims have now been moderated (but see the remarks below).

We thank the reviewer for recognizing that our framework is sufficiently solid to enable concrete numerical threshold estimates in future work. As the primary contribution of our paper is to establish the existence of a fault-tolerant threshold against a general class of CV noise, detailed and improved numerical threshold estimates are left for follow-up studies. In response to the reviewer's comments, we have further revised the manuscript to fully address the points raised, as detailed below.

The newly introduced numerical values in Fig. 5 and the associated discussion for states is derived using an infinite-energy-state, and it is a priori unclear if such bounds have much meaning if translated to more realistic states. I appreciate that the authors mention this caveat prominently, at least. (The discussion in the supplementary material does not fully work out the desired tradeoff, see p. 37.)

We admit that our way of evaluating the noise strength for a state preparation is somewhat indirect and has a caveat. To simplify the numerical calculation, we evaluated the fidelity to an infinite-energy state in Fig. 5, which may lead to an optimistic bound. By evaluating the fidelity or the trace distance between this infinite-energy state and a finite-energy state given in p. 37 and applying the triangle inequality, we could obtain a rigorous bound on the noise strength with a finite energy, which we were not able to perform explicitly due to a lack of numerical resources. We have added the following explanation in p. 37 for readers to carry out the numerical evaluation in principle.

The estimate of the noise strength given in Fig. 5 in Methods is thus optimistic since it computes the trace distance to an infinite-energy state $\|\hat{\Pi}_s |\bar{0}_{\sigma^2}^{\text{app}}\rangle\|^{-1/2} \hat{\Pi}_s |\bar{0}_{\sigma^2}^{\text{app}}\rangle$. The residual term to be evaluated is given by the trace distance between $\|\hat{A}_s^\delta |\bar{0}_{\sigma^2}^{\text{app}}\rangle\|^{-1/2} \hat{A}_s^\delta |\bar{0}_{\sigma^2}^{\text{app}}\rangle$ and $\|\hat{\Pi}_s |\bar{0}_{\sigma^2}^{\text{app}}\rangle\|^{-1/2} \hat{\Pi}_s |\bar{0}_{\sigma^2}^{\text{app}}\rangle$, which may be roughly proportional to δ when $\delta \ll 1$. How small δ can be depends on how large one allows for E_{prep} . We leave a tight evaluation of the noise strength, i.e., the evaluation of the trace distance to $\|\hat{A}_s^\delta |\bar{0}_{\sigma^2}^{\text{app}}\rangle\|^{-1/2} \hat{A}_s^\delta |\bar{0}_{\sigma^2}^{\text{app}}\rangle$, for future work.

I understand the point (now explained in more detail) that a bound with $s = 0$ for noise channels trivially implies one for $s \neq 0$. But this does not make full use of the introduced framework (which presumably could provide better bounds), which is somewhat unfortunate.

Although the value of s can be traded off between different locations in a GKP ExRec, it is definitely good if we have a nontrivial bound on the noise strength with $s \neq 0$ in general. We leave the problem of deriving such a nontrivial bound either analytically or numerically for future work, as mentioned in Sec. A 4.

I appreciate the elimination of the term “explicit bound” in places where no explicit bound was known or provided (following the established terminology that an explicit bound is an analytical formula depending on the given parameters). However, I do not fully understand what is meant by the newly introduced phrasing “with clarification of an upper bound on the noise strength” (used in the abstract and also later in the paper). It is quite unclear, in my opinion, what “clarification” means in such a phrase—this is not standard terminology. Perhaps this can be omitted altogether?

Following the reviewer’s advice, we removed the word “clarification” and fixed the terminology in the revised manuscript.

One additional comment: To clarify the meaning of the GKP *-decoder etc, it would be good to include a reference to <https://arxiv.org/abs/0904.2557> where the meaning of such a decoder and the need is discussed in detail.

In the previous manuscript, we cited the paper in the Supplementary Information but not in the main text; in the revised manuscript, we added it to the main text when introducing the GKP *-decoder.